

# EcH$_2$O-iso 1.0: Water isotopes and age tracking in a process-based, distributed ecohydrological model

Sylvain Kuppel[1], Doerthe Tetzlaff[1,2,3], Marco P. Maneta[4], and Chris Soulsby[1,2]

[1]Northern Rivers Institute, University of Aberdeen, Aberdeen, AB24 3UF, United Kingdom.
[2]Leibniz Institute of Freshwater Ecology and Inland Fisheries, Berlin, 12587, Germany.
[3]Department of Geography, Humboldt University Berlin, Berlin, 10099, Germany.
[4]Geosciences Department, University of Montana, Missoula, MT 59812-1296, USA

**Correspondence:** S. Kuppel (sylvain.kuppel@abdn.ac.uk)

**Abstract.** We introduce EcH$_2$O-iso, a new development of the physically-based, fully-distributed ecohydrological model EcH$_2$O where the tracking of water isotopic tracers ($^2$H and $^{18}$O) and age has been incorporated. EcH$_2$O-iso is evaluated at a montane, low-energy experimental catchment in eastern Scotland using 16 independent isotope time series from various landscape positions and compartments; encompassing soil water, groundwater, stream water, and plant xylem. We find a good

model-observation match in most cases, despite having only calibrated the model using hydrometric data and energy fluxes. These results provide further validation of the physical basis of the model for successfully capturing catchment hydrological functioning, both in terms of the celerity in energy propagation (e.g. runoff generation under prevailing hydraulic gradients) and flow velocities of water molecules (e.g., in consistent tracer concentrations at given locations and times). We also show that the spatially-distributed formulation of EcH$_2$O-iso provides a powerful tool for quantitatively linking water stores and fluxes

with spatio-temporal patterns of isotopes ratios and water ages. Finally, our study highlights some model development and benchmarking needs, refined using isotope-based calibration, for hypothesis testing and improved simulations of catchment dynamics that is transferable beyond the catchment landscape studied here.

## 1 Introduction

Before being evaporated to the atmosphere or routed to the oceans, continental precipitation transits in soils, plants, aquifers,
and rivers. All these pathways in the "critical zone" (National Research Council, 2012) shape the coupling between hydrology and biogeochemistry, and impose controls on many ecological and geomorphological processes. In turn, these interactions determine the partitioning of water trajectories between storage, bypass, mixing, recharge and evapotranspiration (Brooks et al., 2015). In this respect, conservative tracers such as stable water isotopes ($^1$H, $^2$H, $^{16}$O, and $^{18}$O) represent a useful "water fingerprinting" tool to research these mechanisms due to the process-dependent asymmetrical dynamics of heavier and
lighter isotopes. Combined with a quantification of water flux rates and storage dynamics – either measured or modelled –, characterizing isotopic composition provides powerful insights into water pathways at scales ranging from the pedon (Sprenger et al., 2018) to the catchment landscape (McGuire and McDonnell, 2006; Birkel and Soulsby, 2015), and even to global estimates of terrestrial water flux partitioning (Good et al., 2015). Furthermore, tracers have been of particular importance in





understanding catchment functioning, as they highlight pore velocities of water molecules (i.e., how fast does a given parcel of water move) in a way that distinguishes this from the celerity (i.e., how fast energy propagates via the hydraulic gradient) of the rainfall-runoff response (McDonnell and Beven, 2014).

Historically, isotopic transport models were initially developed at the plot scale ($\sim$1-100 m$^2$) to represent 1-D isotope
transfers in the soil profile and at the surface-atmosphere interface (Mathieu and Bariac, 1996; Melayah et al., 1996; Braud et al., 2005, 2009; Haverd and Cuntz, 2010; Soderberg et al., 2012; Sprenger et al., 2018). Process-based simulation of isotopic trajectories has also been considered in larger-scale studies using land surface models (Haverd et al., 2011; Henderson-Sellers, 2006) where couplings with atmospheric isotopic circulation can be captured (Haese et al., 2013; Risi et al., 2016; Wong et al., 2017). While the simulation of energy budgets and biogeochemical cycles is increasingly detailed in land surface models, the
hydrology has, however, remained somewhat simplistic. These shortcomings in explicitly taking into account lateral transfers as overland flow, shallow and deeper subsurface flows and channel routing (Fan, 2015) make it difficult to characterise the role of cascading downstream water redistribution in the spatial patterns of catchment functioning.

In parallel, isotopes have been used to explore water velocities, travel times and ages in catchments using analytical and conceptual models (e.g., Neal et al., 1988; Barnes and Bonell, 1996; Weiler et al., 2003; Sayama and McDonnell, 2009; Birkel
et al., 2015; McGuire and McDonnell, 2015). These numerical tools allow testing hypotheses regarding how catchment storage relates to hydrological fluxes via mixing (or the relative absence thereof), and extending insights to spatio-temporal scales and variables inaccessible to current observation methods. An example of the latter is the estimation of water age, for which such models hold great promise (Dunn et al., 2007; McGuire and McDonnell, 2006; Sayama and McDonnell, 2009), with a more recent focus on the statistical properties of water transit time with time-varying and/or spatially-distributed conceptualizations
(Botter et al., 2010; Birkel et al., 2012; Heidbüchel et al., 2012; Harman, 2015; Rinaldo et al., 2015; Benettin et al., 2017; Hesse et al., 2017). Additionally, the distinct information content of tracer observations, compared to more traditional hydrometric data, dictates that the integration of the two offers a strong hypothesis-testing framework for catchment model development (Uhlenbrook and Sieber, 2005; Fenicia et al., 2008; McDonnell and Beven, 2014). This opportunity is reinforced by decreasing costs of stable isotope analysis, now allowing for collection of daily (or more frequent) time series over several years (Kirchner
and Neal, 2013) to inform simulations.

Yet, applications of a velocity-celerity framework in model-data fusion for catchment-scale hydrology remains relatively rare (Birkel and Soulsby, 2015). Such studies are urgently needed at this scale where the emphasis is mainly on the characterization of water pathways from precipitation to streamflow generation and/or evaporative losses. Recent efforts have nonetheless provided insights, either into whole-catchment dynamics with conceptual rainfall-runoff models (Birkel et al., 2011; Stadnyk
et al., 2013; Hrachowitz et al., 2013; van Huijgevoort et al., 2016; Smith et al., 2016; Ala-aho et al., 2017; Knighton et al., 2017); or at finer detail using process-based 2-D hillslope models (Windhorst et al., 2014). We argue that extending tracer-aided approaches to physically-based models could resolve both intra- and whole-catchment dynamics of stable water isotopes and bridge perspectives at multiple and process-specific scales, as largely shown in hydrometric-based studies (e.g., Endrizzi et al., 2014; Pierini et al., 2014; Niu and Phanikumar, 2015; Manoli et al., 2017). This process-oriented characterisation could also
include non-conservative isotope behaviour such as evaporative fractionation, whereby water with lighter isotopes ($^1$H and $^{16}$O)





preferentially evaporates (Gat, 1996), and whose impact on downstream water signatures has been highlighted even in energy-limited landscapes (Sprenger et al., 2017a). Birkel et al. (2014) and Knighton et al. (2017) are amongst the rare attempts to include fractionation in catchment-scale studies, albeit with conceptual rainfall-runoff models. Investigation of internal catchment heterogeneity, marked in some geographical settings (Tetzlaff et al., 2013), is facilitated by spatially-distributed

resolutions of the catchment domain. In previous tracer-aided catchment modelling however, this aspect is either indirectly considered – e.g., a semi-distributed separation of non-saturated/saturated domains (Birkel et al., 2015) – or simply absent. Where spatial distribution has been taken into account in the model structure (van Huijgevoort et al., 2016; Ala-aho et al., 2017), fractionation processes were not included.

Finally, plants dynamically modulate evaporative losses (ET) – green water, *sensu* Falkenmark and Rockström (2006) –

in the landscape water balance. This crucially drives the partitioning between soil evaporation ($E_s$), evaporation of canopy-intercepted water ($E_c$), and plant transpiration (T). The two former pathways can result in evaporative fractionation, and root uptake for transpiration is usually considered non-fractionating (e.g., Wershaw et al., 1966; Dawson and Ehleringer, 1991; Harwood et al., 1999), although whether this is the case has recently been subject of debate (Lin and da SL Sternberg, 1993; Zhao et al., 2016; Vargas et al., 2017). While these different isotopic dynamics are of key importance in disentangling ecohydrolog-

ical couplings in tracer-aided modelling, previous approaches generally lack a process-based conceptualisation of vegetation. Knighton et al. (2017) separately distinguished T from other ET components in catchment-wide isotopic model-data fusion. However, their spatially-lumped approach was parsimonious, using empirical partitioning of potential evapotranspiration which has high uncertainty in natural ecosystems (Kool et al., 2014).

Here, we implement isotope and age tracking in the physically-based, fully-distributed model EcH$_2$O (Maneta and Silver-

man, 2013). Notably, this model separately solves the energy balance at the top of the canopy and at the soil surface, allowing a process-based separation of soil evaporation, transpiration, and canopy evaporation. The novel isotopic and age tracking is designed in a fashion directly consistent with the original model structure, assuming full mixing in each model compartment, and with very limited empirical parameterization. The critical conceptualisation of evaporation fractionation uses the well-known Craig-Gordon approach (Craig and Gordon, 1965). The new tracer-enhanced model (EcH$_2$O-iso) is tested in a small

low-energy montane catchment where, in addition to long-term, high resolution isotopic datasets for rainfall and runoff ($^2$H and $^{18}$O), we assess the spatio-temporal variation of model-data agreement in soil water, groundwater, and plant xylem. Crucially, no isotopic calibration is conducted. The site has previously been modelled aplying EcH$_2$O for calibration, using multiple datasets of ecohydrological fluxes and storage variables (Kuppel et al., 2018). We ask the following questions: *1)* To what extent can a hydrometrically-calibrated, physically-based hydrologic model correctly reproduce internal catchment dynamics of

isotopes? *2)* What are the limitations of these isotopic simulations? Do they relate to the physics and/or to mixing assumptions? *3)* What are the implications and opportunities for simulating spatio-temporal patterns of isotopes and water ages?





## 2 Model description

### 2.1 Presentation of the EcH₂O model

The ecohydrological model EcH$_2$O combines a land surface module for calculating vertical energy balances (canopy and understory), with a kinematic-wave-based scheme for lateral and vertical water transfers, while vegetation productivity and growth is derived from plant transpiration (Maneta and Silverman, 2013). Energy fluxes, water fluxes and storage, and vegetation state are explicitly coupled to capture the feedbacks between ecosystem productivity, hydrology and climate, at time steps larger or equal to that of the meteorological inputs (precipitation P, incoming longwave and shortwave radiation, air temperature (maximum, average, and minimum), relative humidity, and wind speed). A comprehensive description of the model can be found in (Maneta and Silverman, 2013), and subsequent developments in Lozano-Parra et al. (2014) and Kuppel et al. (2018).

We provide here a brief step-wise overview, focused on the different hydrological compartments and transfers simulated in EcH$_2$O at the grid cell level (Fig. 1). For each vegetation cover present in a grid cell, a linear bucket approach is used for canopy interception. The capacity-excess P (i.e., throughfall) is partitioned between liquid and snow components using for each time step the minimum and maximum air temperature, together with a snow-rain temperature threshold. The canopy energy balance then separately yields plant transpiration from canopy conductance, and evaporation of intercepted water. Infiltration of liquid throughfall in the topsoil is computed using a Green and Ampt approximation of Richard's equation. Subsequent soil water content above field capacity (gravitational water) percolates to the underlying soil layers, with fixed bedrock seepage – out of the system – as a lower boundary condition (Fig. 1). Soil evaporation (limited to the topsoil) and snowmelt under each vegetation type are calculated by solving the energy balance at the surface (soil or snowpack). Following a local drainage direction derived from the input elevation map, lateral water routing is simulated at three levels: in the deepest soil layer, groundwater seeps in the channel (if present) while the remainder is transferred laterally using a 1D kinematic wave, and can result in saturation return flow in downstream cells. All remaining ponded water becomes overland flow; reinfiltrating further downstream or running off until it reaches an outlet or a cell within the stream network; stream water routing is also computed within a 1D kinematic wave approximation.

### 2.2 Implementation of isotopic and age mixing

The conceptualization of water mixing equally applies for all the tracked quantities implemented in the model (isotopes and age), so that a generic notation $C$ is in this section used to designate both isotopic tracer composition ($^2$H and $^{18}$O) and water age. The only specificities of isotope dynamics in EcH$_2$O-iso relates to fractionation (see Sect. 2.3), while precipitation inputs have a fixed age of zero and the water age in all compartments is incremented at the end of each simulation time step by the length of the latter. The delta notation ($\delta$) for isotopic composition quantifies, for a given water sample, the difference in the mass ratio of heavy to light isotopes (R) as compared to the Vienna Standard Mean Ocean Water (VSMOW): $\delta = \left( \frac{R_{sample}}{R_{VSMOW}} - 1 \right) \times 10^3$





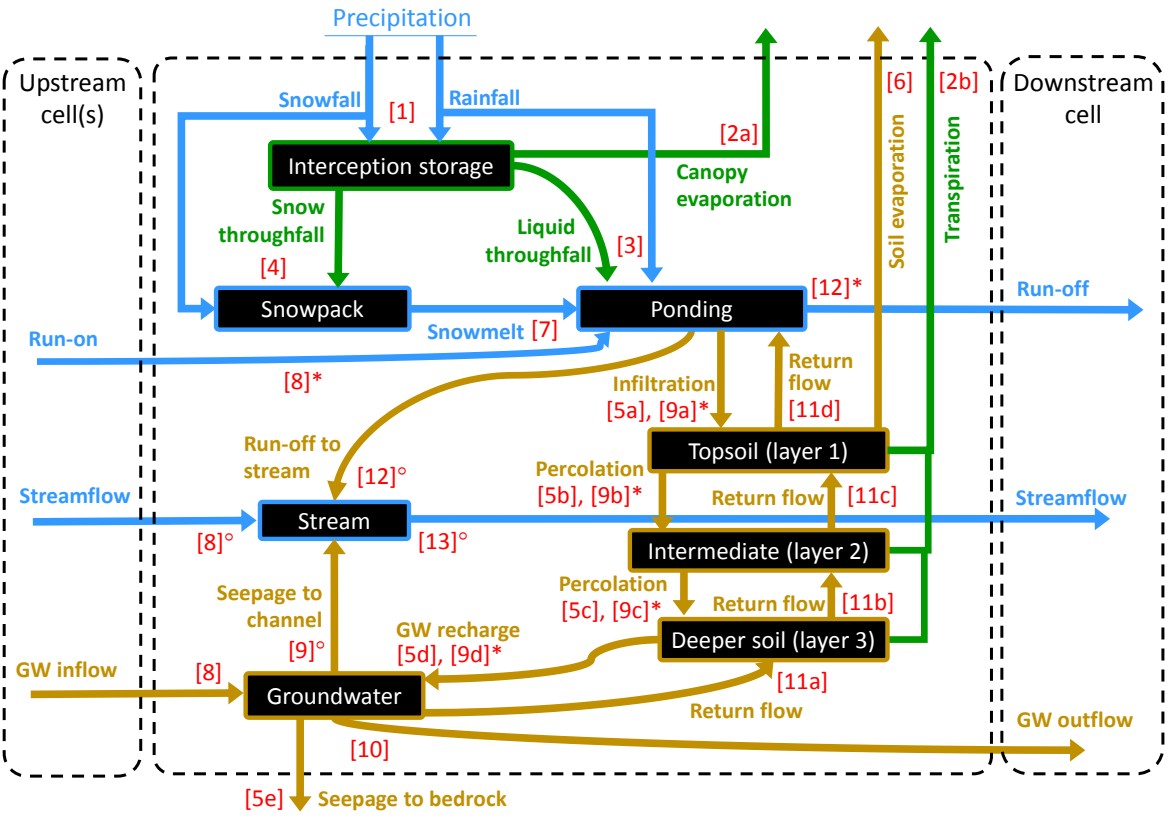

**Figure 1.** Water compartments (black rectangles) and fluxes (coloured arrows) as represented in EcH$_2$O, with the numbers between brackets reflecting the sequence of calculation within a time step. Note that water routing (steps [8] to [13]) differs between cells where a stream is present (◦) or not (∗).

. First, the instantaneous mass balance for water signature is:

$$\frac{d(V_{res}C_{res})}{dt} = \sum_{k=1}^{N_{in}} q_{in,k}C_{in,k} - q_{out}C_{res} \tag{1}$$

where $V_{res}$ and $C_{res}$ are, respectively, the volume and signature ($\delta^2 H$, $\delta^{18}O$, or age) of the water in the reservoir, $t$ is time, $q_{out}$ is the flux of water exiting the reservoir, and $q_{in,k}$ and $C_{in,k}$ are, respectively, the flux and signature of water entering the

reservoir from each the $N_{in}$ adjacent upstream locations. An implicit first-order finite-difference scheme is used to compute mixing during a given time interval $\Delta t$ :

$$V_{res}^{t+\Delta t}C_{res}^{t+\Delta t} - V_{res}^{t}C_{res}^{t} = \left(\sum_{k=1}^{N_{in}} q_{in,k}C_{in,k}^{t+\Delta t} - q_{out}C_{res}^{t+\Delta t}\right) \cdot \Delta t, \tag{2}$$





where $V_{res}^{t+\Delta t}$ and $C_{res}^{t+\Delta t}$ are, respectively, the volume and water signature in the reservoir after mixing, $V_{res}^t$ and $C_{res}^t$ are the volume and water signature in the reservoir before mixing, $C_{in,k}^{t+\Delta t}$ is the signature of the $k$-th input source after mixing in the latter (following the implicit approach), and $\Delta t$ is the time step. Replacing $V_{res}^{t+\Delta t}$ by $V_{res}^t + \left( \sum_{k=1}^{N_{in}} q_{in,k} - q_{out} \right) \cdot \Delta t$ in Eq. (2) finally yields:

$$C_{res}^{t+\Delta t} = \frac{V_{res}^t C_{res}^t + \left( \sum_{k=1}^{N_{in}} q_{in,k} C_{in,k}^{t+\Delta t} \right) \Delta t}{V_{res}^t + \left( \sum_{k=1}^{N_{in}} q_{in,k} \right) \Delta t}. \tag{3}$$

In practice, Eq. (3) is applied in EcH$_2$O-iso at every sub-time step where water transfers are computed, in the sequence shown in Fig. 1. Note that $C_{res}^{t+\Delta t}$ in Eq. (3) only depends on the magnitude of the summed incoming flux $\sum_{k=1}^{N_{in}} q_{in,k}$ . Flow to the downstream cell is fully mixed – right-hand terms of Eq. (2). Full mixing was used as a simplifying approximation because this model is to be first evaluated in a wet environment (Tetzlaff et al., 2014; Sprenger et al., 2017a) with relatively long time steps (i.e. daily, see Sect. 3.3). One exception to immediate mixing is the snowpack, where the snowmelt flux $\left( q_{out}^{melt} \right)$

is assumed to tap first into the snow throughfall of the same day ($q_{in}^{snow}$) if present, before mobilizing older snow, fully mixed in the snowpack. Consequently, the signatures of the snowpack ($C_{pack}^{t+\Delta t}$) and snowmelt water ($C_{melt}^{t+\Delta t}$) which goes into the surface reservoir in EcH$_2$O-iso at step 7 (Fig. 1) are calculated as follows:

$$C_{pack}^{t+\Delta t} = \frac{V_{pack}^t C_{pack}^t + max\left(0, q_{in}^{snow} - q_{out}^{melt}\right) C_{rain}^{t+\Delta t} \Delta t}{V_{pack}^t + max\left(0, q_{in}^{snow} - q_{out}^{melt}\right) \Delta t} \tag{4}$$

$$C_{melt}^{t+\Delta t} = \frac{max\left(0, q_{out}^{melt} - q_{in}^{snow}\right) C_{pack}^{t+\Delta t} + q_{in}^{snow} C_{rain}^{t+\Delta t}}{max\left(q_{in}^{snow}, q_{out}^{melt}\right)} \tag{5}$$

    No spill-over of canopy-intercepted water is simulated in this bucket-type approach of EcH$_2$O. As a result, intercepted water evaporates from the canopy and does not interact with the surface/subsurface. Therefore, throughfall water (liquid and snow) is assumed to have the isotopic composition of same-time-step precipitation and age zero. This simplification is reasonable for our study site where vegetation interception has only a trivial effect on the isotopic partitioning of rainwater (Soulsby et al.,

2017), yet further developments could be implemented for model application in different eco-climatic settings.

    Finally, transpiration is considered as a non-fractionating process. This is based on previous work (Wershaw et al., 1966; Dawson and Ehleringer, 1991; Harwood et al., 1999), and the fact that non-steady state effects cancel out at the daily time-step (Farquhar et al., 2007). However, this simple conceptualisation is increasingly questioned (Lin and da SL Sternberg, 1993; Zhao et al., 2016; Vargas et al., 2017), and the implications for our study will be discussed later. Here, during the canopy

energy balance (step [2] in Fig. 1), the signature of transpired water $C_T$ is taken as the weighted sum of the signature in the three soil layers:

$$C_T = f_{L1}C_{soilL1} + f_{L2}C_{soilL2} + f_{L3}C_{soilL3}, \tag{6}$$

    where $f_{L1}$ , $f_{L2}$ , and $f_{L3}$ are the respective fractions of roots in each layer, as described in Eq. (A8) in Kuppel et al. (2018).





### 2.3  Isotopic fractionation from soil evaporation

The change in isotopic composition of the first soil layer during soil evaporation (step [6] in Fig. 1) is simulated using the Craig-Gordon model Craig and Gordon (1965); Gat (1995), without any empirical parameterization specific to our study. In this section, generically refers to the standardized isotopic ratio in for either $^2$H or $^{18}$O. For each time step $t$:

$$\delta_{soilL1}^{t+\Delta t} = \delta^* - \left(\delta^* - \delta_{soilL1}^t\right) f^m, \tag{7}$$

where $f$ is the remaining fraction of water after evaporation ( $f = V_{soilL1}^{t+\Delta t}/V_{soil}^t$ ), while $\delta^*$ is the limiting isotopic composition given the local atmospheric conditions in ‰(Gat and Levy, 1978) and $m$ is the dimensionless enrichment slope (Welhan and Fritz, 1977; Allison and Leaney, 1982). Their formulation is generalized following Good et al. (2014):

$$\delta^* = \frac{h_a\delta_a + h_s \cdot \varepsilon^+ + \varepsilon_k}{h_a - (h_s \cdot \varepsilon^+ + \varepsilon_k) \cdot 10^{-3}} \tag{8}$$

$$m = \frac{h_a - (h_s \cdot \varepsilon^+ + \varepsilon_k) \cdot 10^{-3}}{h_s - h_a + \varepsilon_k \cdot 10^{-3}} \tag{9}$$

The different terms in Eqs. (8) and (9) are sequentially defined as follows:

- $\delta_a$ is the stable isotope composition of the ambient air moisture in ‰, derived from that of the precipitation by assuming isotopic equilibrium Gat (1995); Gibson and Reid (2014):

$$\delta_a = \frac{\delta_{rain} - \varepsilon^+}{\alpha^+} \tag{10}$$

- $\varepsilon^+$ is a factor (in ‰) derived from the equilibrium fractionation $\alpha^+$ of water between the liquid and vapour phases (Skrzypek et al., 2015):

$$\varepsilon^+ = \left(1 - 1/\alpha^+\right) \cdot 10^3 \approx \left(\alpha^+ - 1\right) \cdot 10^3, \tag{11}$$

with $\alpha^+$ taken as temperature-dependent following Horita and Wesolowski (1994), here using the air temperature $T_a$ :

$$10^3 \cdot \ln\alpha_{^2H}^+ = \frac{1158.8}{10^9} \cdot T_a^3 - \frac{1620.1}{10^6} \cdot T_a^2 - \frac{794.84}{10^3} \cdot T_a - 161.04 + \frac{2.9992}{T_a^3} \cdot 10^9 \tag{12}$$

$$10^3 \cdot \ln\alpha_{^{18}O}^+ = -7.685 + \frac{6.7123}{T_a} \cdot 10^3 - \frac{1.6664}{T_a^2} \cdot 10^6 + \frac{0.35041}{T_a^3} \cdot 10^9 \tag{13}$$

- $\varepsilon_k$ accounts for the diffusion-controlled fractionation in air (Craig and Gordon, 1965):

$$\varepsilon_k = (h_s - h_a) \cdot n \cdot \left(1 - \frac{D_i}{D}\right), \tag{14}$$

where $D_i/D$ is the diffusivity ratio of the gaseous water molecules bearing an isotope $i$ to that of lighter isotopic water. We use literature values given by Vogt (1976), as suggested in Horita et al. (2008): 0.9877 for $D_{^1H^2HO}/D_{H_2O}$ and 0.9859 for $D_{H_2^{18}O}/D_{H_2^{16}O}$ .





- $n$ translates the dominant mode of transport of water molecule at the surface. We adopted a time-varying formulation taking into account soil water content $\theta$ (Braud et al., 2005; Mathieu and Bariac, 1996):

$$n = 1 - 0.5 \cdot \frac{(\theta - \theta_r)}{\phi - \theta_r} \tag{15}$$

where $\phi$ and $\theta_r$ are, respectively, the soil porosity and residual water content. $n$ increases from 0.5 in a saturated soil to 1 for a dry soil where diffusion is the dominant mode of transport.

- $h_a$ is the relative humidity of the atmosphere (measured at the weather stations, see Sect. 3.2) after being normalized to the saturated vapor pressure $e^*$ at the soil surface (Gat, 1995):

$$h_a = h_{a,measured} \cdot \frac{e^*(T_a)}{e^*(T_s)}, \tag{16}$$

where the surface temperature $T_s$ is given at each time step from solving the surface energy balance equation (Maneta and Silverman, 2013).

- Finally, $h_s$ is the relative humidity of the air of the soil pores, following the formulation of soil evaporation flux $E$ in EcH$_2$O (Eqs. 9-10 in Maneta and Silverman, 2013):

$$h_s = \beta + (1 - \beta) \cdot h_a, \tag{17}$$

where $\beta$ is adjusted as a growing function of the volumetric water content $\theta$, equal to 1 whenever $\theta$ is superior or equal to field capacity $\theta_{fc}$ (Lee and Pielke, 1992):

$$\beta = min\left(1, \frac{1}{4}\left[1 - cos\frac{\theta}{\theta_{fc}}\pi\right]^2\right). \tag{18}$$

## 3 Data and methods

### 3.1 Study site

Simulations were conducted for the Bruntland Burn (BB) catchment in the Scottish Highlands (57°8'N 3°20'W) (Fig. 2a-b). It is a small (3.2 km$^2$) headwater catchment of the Dee, a major Scottish river providing freshwater resources for 250,000 people in the Aberdeen urban area, having EU conservation designations, and hosting ecosystem services (e.g., Atlantic salmon fishery). Annual precipitation averages around 1000 mm, with a mild seasonal cycle (Fig 2c). The water balance is energy-limited, given the northern latitude, with 400 mm of annual potential evapotranspiration (PET) with pronounced seasonality in daily losses: from 0.5 mm in winter to 4 mm in summer. Mean annual temperature is 7°C and no monthly-averaged temperatures fall below 0°C, the climate qualifies as temperate to boreal oceanic; less than 5% of precipitation occurs as snowfall.

The topography of the BB reflects glacier retreat, with elevation ranging from 220 m.a.s.l in the wide valley bottom to 560 m.a.s.l on above the steeper slopes (Fig. 2a). Glacial drift deposits cover 60-70% of the catchment bedrock (granite, schist and





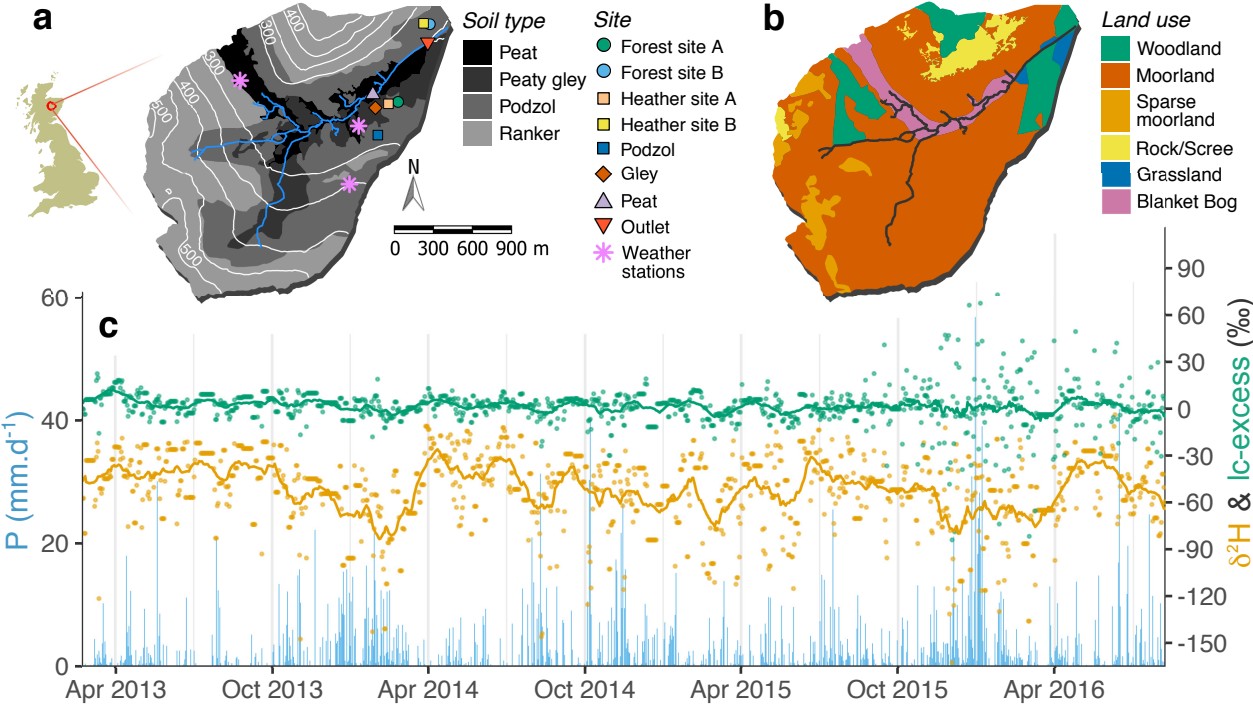

**Figure 2.** Bruntland Burn catchment characteristics, showing **(a)** topography, soil cover as derived from the Hydrology of Soil Types (HOST) classification types, stream network, and measurements sites locations, and **(b)** land use type. **(c)** Time series of measured precipitation amount (blue bars, daily) and isotopic signatures, $\delta^2$H (orange) and lc-excess (green), showing daily values (dots) and the 30-day running mean (solid lines).

other meta-sediments) and forms the dominant soil parent material (Soulsby et al., 2007). Mostly saturated, these deposits are important reservoirs of groundwater, sustaining base flows in the stream and maintaining persistent wet conditions across the

5    valley bottom (Soulsby et al., 2016). Thin regosols (rankers) dominate the pedology of the catchment above 400 m.a.s.l., where drift deposits are marginal (Fig. 2a). Freely-draining shallow podzols (<0.7 m deep) dominate steeper hillslopes, overlying moraines and marginal ice deposits. Finally, deep (>1 m) soils with high organic matter content (histosols: peat and gley) characterize the riparian area (Fig. 2a). The histosols are saturated most of the time, so that rainfall events generate runoff mostly via rapid saturation overland flow, with a surface connectivity in the podzols limited to the wettest periods (Tetzlaff

10    et al., 2014). Spatial patterns of land cover reflect these hydropedological units (Fig. 2b). Heather shrublands (*Calluna vulgaris* and *Erica* spp.) are the dominant cover over podzols and rankers. Such a land use results from red deer (*Cervus elaphus*) and sheep overgrazing, at the expense of naturally-occurring Scots pine trees (*Pinus sylvestris L.*), which are now mostly found in the steep sections of the northern hillslopes and in the plantation areas neighbouring the stream outlet. Finally, grasses (*Molinia caerulea*) cover the riparian gley soils, while the peat is dominated by bog mosses (*Sphagnum spp.*).





**Table 1.** Local datasets used in this study, grouped by location and purpose: model evaluation (■), model calibration (▲), and model inputs (♦). For soil isotopes, [a] and [b] respectively indicate suction-lysimetric sampling (2013) and direct equilibration from soil sampling (2015-2016). Other notation: Srf – surface water, GW – groundwater, P – precipitation, SWC – soil water content, Tp – transpiration, NR – net radiation, ∗ – relative air humidity, precipitation, air temperature, and wind speed, ● – synoptic collection campaign at 92 to 94 locations (see text).

| Locations | Water isotopes | | | | | | Water fluxes & stores | | | Meteorology | |
| | Soil | Srf | GW | Xylem | Stream | P | SWC | Pine Tp | Discharge | NR | Other∗ |
|---|---|---|---|---|---|---|---|---|---|---|---|
| Forest site A | ■ [a,b] | | | ■ | | | | ▲ | | | |
| Forest site B | ■ [b] | | | ■ | | | ▲ | ▲ | | | |
| Heather site A | ■ [b] | | | ■ | | | | | | | |
| Heather site B | ■ [b] | | | ■ | | | | | | | |
| Podzol | ■ [a] | | ■ | | | | ▲ | | | | |
| Gley | ■ [a] | | ■ | | | | ▲ | | | | |
| Peat | ■ [a] | | ■ | | | | ▲ | | | | |
| Riparian area ● | | ■ | | | ■ | | | | | | |
| Outlet | | | | | ■ | ♦ | | | ▲ | | |
| Weather stations | | | | | | | | | | ▲ | ♦ |

## 3.2 Datasets

We used the wealth of diverse and often multi-year time series available at different locations in the BB catchment (Fig. 2a). These measurements capture numerous ecohydrological processes and observables, used either for model inputs, or calibration/evaluation of simulations (Table 1). A brief description follows.

### 3.2.1 Isotopic measurements

At the catchment outlet, rainfall and stream water have been sampled daily for isotope analysis from June 2011 to the present, providing an isotope time series of unusual high frequency and longevity. Samples have been collected using an ISCO 3700 automatic sampler (Teledyne Isco, Lincoln, USA). The auto-sampler bottles were emptied at fortnightly frequency or higher, while paraffin was added to each bottle to prevent evaporation.

Stable isotope measurements in the soil fall into two categories, differing in the sampling method and time period. Between 2011 and 2013, soil water was extracted at 0.1, 0.3, and 0.5 m depth at four locations: Peat, Gley, and Podzol sites (weekly), and Forest site A (fortnightly) (Fig. 2a). Since MacroRhizon suction lysimeters were used (Rhizosphere Research Products, Wageningen, Netherlands) (Tetzlaff et al., 2014), isotopic characterization represents the mobile water (Sprenger et al., 2015). From September 2015 to August 2016, near-monthly soil water sampling was carried out at four sites (Forest sites A and B, and Heather sites A and B) using soil samples collected with a spade from four layers (0-0.05, 0.05-0.1, 0.10.15, and 015-0.2.



30 m) with five replicates for each. Isotopic analysis followed on water extracted by the direct equilibration method (Wassenaar et al., 2008), thus fully accounting for bulk pore water, as described by Sprenger et al. (2017a). Conceptually, the lysimeters can be viewed as sampling the "fast domain" of soil water held under low tension, whilst direct equilibration characterises the "bulk" soil water which also includes the "slow domain" of water held under higher tensions.

Groundwater samples were collected monthly between August 2015 and September 2016, at four wells (>1.6 m) covering a representative transect from the hillslope to valley bottom (Scheliga et al., 2017) encompassing the main hydropedological units; Peat (2 wells), Gley and Podzol sites (Fig. 2a). Vegetation xylem water was collected between Autumn 2015 and Spring-Summer 2016, using cryogenic extraction from Scots Pine xylem cores at 1.5 m height (Forest sites A and B) and from heather

twigs (Heather sites A and B) (Fig. 2a). Sampling was made at near-monthly resolution (n = 7) with five replicas for each extraction (Geris et al., 2017). We also used isotopic measurements from a synoptic sampling campaign conducted in the drainage network of pools and channels across the valley bottom of the Bruntland Burn on 20[th] February (92 locations) and 24[th] May (94 locations) of the year 2013, covering contrasting catchment wetness states. On those days, water was also sampled along the perennial stream network at 10 locations (Lessels et al., 2016).

Air tight vial were used to store all water samples, which were kept refrigerated until they were analysed. The soil samples were equilibrated and extracted water analysed within a week of collection (Sprenger et al., 2017a). In both cases, stable isotopic composition was determined using Los Gatos laser isotope spectrometers (DLT-100 and OA-ICSO models; Los Gatos Research, Inc., San Jose, USA), with reported measurement uncertainties of 0.4 and <0.55 ‰($\delta^2$H), and 0.1 and <0.25 ‰($\delta^{18}$O), respectively.

**3.2.2 Hydrometric and meteorological data**

Daily soil moisture data was derived from 15-minute retrievals at four locations: three along the peat-gley-podzol transect presented in Sect. 3.2.1 (Tetzlaff et al., 2014), and in a Scots pine stand (Forest site B, Fig. 2a). Time domain reflectometry (TDR) soil moisture probes (Campbell Scientific, Inc. USA) were located at different depths (0.1, 0.3, and 0.5 m – only 0.1 and 0.3 m in the peat), and replicated $\sim$ 2 m apart. During two growing seasons, Scots pine transpiration was measured at Forest

site A (July – September 2015) and at Forest site B (April – September 2016) (Fig. 2a), by installing Granier-type thermal dissipation sap flow sensors (Dynamax Inc., Houston, USA) on 10 and 14 trees, respectively. Depending on its stem diameter (10 to 32 cm), each tree had 2 to 4 sensors. At the end of each study period, incremental wood core sampling in surrounding trees provided sapwood-area-to-tree-diameter relationships, used to derive stand-scale transpiration estimates (Wang et al., 2017a), which were then daily averaged. At the catchment outlet (Fig. 2a), 15-minute stage height records (Odyssey capacitance probe,

Christchurch, New Zealand) were used to generate daily discharge observations, with a rating curve previously calibrated for a stable stream section.

Finally, meteorological observations used as model inputs (P, air temperature $T_a$, relative humidity, and wind speed) and for calibration (net radiation) were primarily measured at the three meteorological stations which were installed at different landscape positions (valley bottom, bog, and hilltop, Fig. 2a) and operated from July 2014. Prior to that period, a square eleva-

tion inverse distance-weighted algorithm was applied to interpolate local precipitation values from five Scottish Environment



Protection Agency (SEPA) rain gauges located within 10 km of the Bruntland Burn catchment Birkel et al. (2011). Daily mean $T_a$, relative humidity and wind speed values were then available from the Centre for Environmental Data Analysis (CEDA) at the Balmoral station ( 5km NW) (Met Office, 2017). The ERA-Interim climate reanalysis dataset (Dee et al., 2011) was used to retrieve daily minimum and maximum $T_a$ (prior to July 2014), and incoming shortwave and longwave radiation (whole study period). Finally, we applied altitudinal effects on P and $T_a$ were accounted for: we applied a 5.5% increase of P every 100 m.a.s.l. (Ala-aho et al., 2017), and a 0.6°C decrease per 100 m.a.s.l, from the moist adiabatic temperature lapse rate (Goody and Yung, 1995).

## 3.3 Model set-up and calibration

The methodology closely follows the approach detailed in Kuppel et al. (2018). Here, we only provide a brief summary and highlight the modifications adopted for this study.

All simulations were performed on daily time steps, at a $100\times 100$ m$^2$ resolution. This choice of coarser grid cells – from $30\times 30$ m$^2$ in Kuppel et al. (2018) – was made to decrease computation time while preserving reasonable spatial variability across the catchment. The simulation period extends from February 2013 to August 2016, for which a continuous record of daily $\delta^2H$ and $\delta^{18}O$ in precipitation input was available (see Sect. 3.2.1). For all simulations a 3-year spin up period was added using the first three years of isotopic and climatic model inputs, as preliminary tests suggested this spin up was enough to remove transient dynamics.

Based on the soil classes defined by the Hydrology of Soil Types (HOST), four hydropedological units were defined (Fig. 2a) (Tetzlaff et al., 2007) to map soil hydrological properties in the modelled domain. Physical soil characteristics relating to the energy balance were considered as uniform across the catchment, similar to Kuppel et al. (2018) (see Table S1 therein). Land cover was divided into five classes, four of them vegetated: Scots pine, heather shrubs, peat moss, and grasslands. From extensive land use mapping (Fig. 2b), the cover fraction of each vegetation type was estimated by combining $1\times 1$ m$^2$-resolution LiDAR canopy cover measurements (Lessels et al., 2016), aerial imagery and typical vegetation patterns in the different soil units (Tetzlaff et al., 2007; Kuppel et al., 2018). In addition, we took into account occurrence of exposed rock by fixing the depth of the first soil layer to 0.001 m to limit soil evaporation wherever the fraction of bare soil was larger than 0.5.

Finally, the calibrated model parameters, and associated sampling ranges, are those presented in Kuppel et al. (2018). The parameter space was sampled using a uniform Monte-Carlo approach. The corresponding 150,000 simulations were constrained by simultaneously using measurements of stream discharge, soil moisture (4 sites), pine transpiration (2 sites) and net radiation (3 sites) over the whole simulations period. For soil moisture observations, a b-spline curve was fitted to the measured profile (to account for non-monotonic variations) on each sampling date, followed by a vertical integration. It enabled a consistent comparison against simulations in each of the upper two hydrological layers of EcH$_2$O (cf. Fig. 1), while profile-averaged values were used for calibration in Kuppel et al. (2018). Constraints were combined in a multi-criteria objective function based on the cumulative distribution functions (CDF) of dataset-specific goodness-of-fit (GOF) (Ala-aho et al., 2017): mean absolute error for stream discharge and root mean square error for all others observations. This method allows retention of model pa-




rameter sets that give most behavioural simulations *simultaneously* across different variables (Kuppel et al., 2018). We retained the 30 "best" of these parameterizations as a testbed for ensemble simulations of stable water isotopes and water age dynamics.

### 3.4 Analysis

Daily, seasonal and inter-annual climate variability results in changing isotopic composition of precipitation inputs. Equilibrium isotopic fractionation processes result in a strong correlation between rainfall $\delta^2H$ and $\delta^{18}O$ across the globe, defining a global meteoric water line (GMWL, Dansgaard, 1964). At the BB catchment, there is a seasonal trend of more enriched values in

summer and depleted in winter (e.g. Fig. 2c). A local meteoric water line (LMWL) was defined, using daily values from February 2013 to August 2016 and weighting by precipitation inputs ($r^2 = 0.96$, p < 0.001):

$$\delta^2H = 7.8 \cdot \delta^{18}O + 4.9. \tag{19}$$

The line-conditioned excess (hereafter, lc-excess) was defined as the residual from the LMWL (Landwehr and Coplen, 2006):

$$lc - excess = \delta^2H - a_{LMWL} \cdot \delta^{18}O - b_{LMWL}, \tag{20}$$

with $a_{\text{LMWL}} = 7.8$, $b_{\text{LMWL}} = 4.9$ ‰(Eq. 19). As oxygen has a higher atomic weight, non-equilibrium fractionation during the liquid-to-vapour phase change will preferentially evaporate (in terms of statistical expectation) $^1H^2H^{16}O$ molecules rather than the heavy isotopologue $^1H_2{}^{18}O$ (Craig et al., 1963). The isotopic signature of a water sample affected by evaporation thus shows negative lc-excess values, as $\delta^{18}O$ in non-evaporated water enriches faster than $\delta^2H$, and plots under the LMWL in

the dual-isotope space (Landwehr et al., 2014). For these reasons, we preferred combining $\delta^2H$ and lc-excess in our analysis (over separately looking at both $\delta^2H$ and $\delta^{18}O$), to simultaneously highlight absolute isotopic dynamics and evaporative fractionation. Note that lc-excess was also preferred over the oft-used deuterium-excess, which translates the deviation of $\delta^2H$ from the GMWL (Dansgaard, 1964). While the two quantities are mathematically similar, lc-excess displays much smaller seasonal dynamics from the near-0 ‰value of precipitation inputs, thus it advantageously allows separation of fractionation

impacts from overall isotopes dynamics (Sprenger et al., 2017a).

Similar to soil moisture observations, measured and simulated isotopic values in the soil are conceptually different: datasets are collected at specific depths (see Sect. 3.2.1), whilst model outputs provide average values for the different hydrological layers (Fig. 1). While original quantities were preserved for temporal analysis of the results, we additionally provided a formal quantification of model-data agreement. To do so, we reconstructed layer-integrated isotopic datasets at each soil sampling

site, following the same interpolation-integration methodology used for soil moisture for computing model-data goodness-of-fit during calibration (Sect. 3.3). Model evaluation then used model-to-data ratio of standard deviations, and model-data Pearson's correlation factor.



## 4  Results

### 4.1  Time series

Seasonal dynamics of soil water isotopes were well captured on the hillslopes, as exemplified at two sites in Fig. 3: one located in the shrub-dominated moorland (Podzol site), the other in a Scots pine plantation (Forest site B), noting that the graphs cover different hydrological years dictated by data availability. Model-data agreement was consistent for $\delta^2$H, keeping in mind that while measurements were depth-specific, simulated values were averaged over the first and second hydrological layers (Fig. 1). As a result of model calibration (see Sect. 3.3), the thickness of the first (topsoil) and second layers span 0.10-0.19 m and 0.02-0.39 m in the simulated podzol soil unit, respectively (not shown). Still, at the podzol site the model captured well the vertical variability $\delta^2$H across the summer of 2013, but overestimated topsoil enrichment during the following winter (Fig. 3a). Lc-excess was generally underestimated in the topsoil there, with negative simulated values indicating evaporative influence generally not found in the data.

generally not found in the data. At Forest site B, both $\delta^2$H and lc-excess dynamics showed modelled ranges consistent with measurements. Note, however, that EcH$_2$O-iso simulated a vertical profile during the winter 2016 with richer $\delta^2$H in the deeper layer, a condition that was only occasionally found in $\delta^2$H measurements (Nov 2015 and Jan 2016). At both sites, the temporal dynamics of soil moisture were well captured by the model (bottom rows), although the observed decrease of moisture content with depth – especially marked at forest site B – was generally not reproduced by the gravity-driven physics of EcH$_2$O.

Isotopic consistency was also found further in the valley bottom, as shown at the peat site in the riparian area (Fig. 4). The bimodal summer $\delta^2$H enrichment measured was well captured in the topsoil layer of the model (thickness: 0.02-0.25 m), as were the mildly-negative lc-excess values. In addition, the weak variability and range of measured $\delta^2$H and lc-excess at greater depths were consistently reproduced. As for the podzol site, we noted in the peat higher peak enrichment values from the model than for the available data. As for other elements of the analysis, we remind here that soil isotopic data was sampled differently

at the three sites: lysimeters extraction was used for the podzol and peat sites, therefore characterizing mobile water in the fast domain, while the direct equilibration analysis conducted at Forest site B effectively applies to bulk water including water held under higher tension. The model essentially gives a bulk isotopic composition of stored water (Sect. 2), which might also explain why results were comparatively better at Forest site B.

We also explored the accuracy of simulated spatial patterns of isotopic signatures in the riparian zone, using two synoptic

sampling surveys of surface water and stream water (Sect. 3.2.1) on separate days in late winter and late spring of 2013 (Fig. 4d-g). A good agreement was found for $\delta^2$H in the main branch of the stream network on both dates, while there was a tendency to overestimate $\delta^2$H values in the northwest part of the riparian area. Model-data fit of lc-excess mostly oscillated between good and a few per mille underestimation, depending on the location. For both $\delta^2$H and lc-excess there was fine-scale spatial variability in the model-data fit, especially marked in late May and in the main channel. This reflects both the spatial

variability of measurements (Lessels et al., 2016; Sprenger et al., 2017b) and the different resolution of sampling ($\sim$ 10 metres intervals) and the much coarser grid of the simulations ($100\times100$ m$^2$).

Figure 5 shows EcH$_2$O-iso's simulation of the isotopic imprint of plant water uptake in the transpiration flux. The isotopic composition of xylem water samples was directly compared to that of root water uptake simulated from the canopy energy





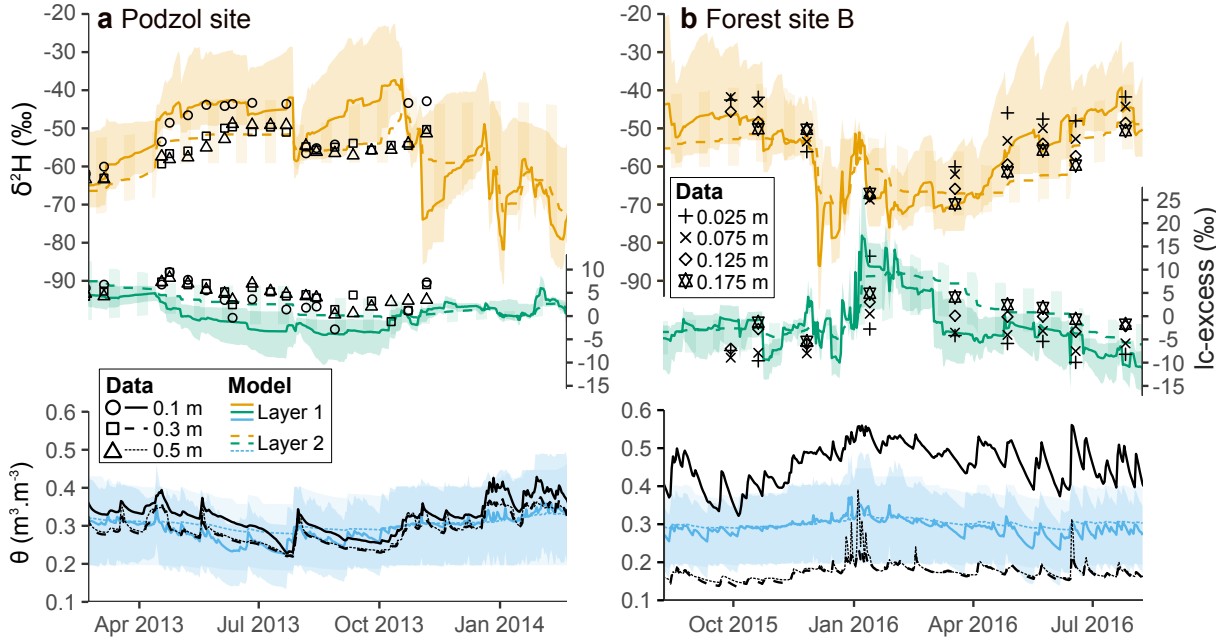

**Figure 3.** Time series isotopic composition ($\delta^2$H – top, and lc-excess – middle) and soil volumetric water (bottom) at two sites located in the hillslopes; **(a)** one dominated by a heather shrub cover and **(b)** the other in a pine-dominated area. Black symbols and lines show measurements at a given depth while colours display the ensemble medians and 90%-intervals of simulations in the two uppermost soil layers.

balance (sub-step [2b] in Fig. 1). At the heather sites, the simulated ranges were consistent from model to data, with an
excellent model-data fit for lc-excess despite the low 90% -spread of simulations outputs (Figs. 5a and 5b). The seasonal cycle of simulated $\delta^2$H conversely seemed opposite to that of xylem samples, which showed gradual enrichment in winter followed by depletion at beginning of the growing season, but the lack of data from January to April limits general seasonal interpretation. At the forest sites, simulation results were very similar, noting that Forest site A corresponds to the same model grid cell as Heather site A (Figs. 5c and 5d). However, the measured isotopic composition in xylem was quite different for Scots Pine compared to the heather, in two ways. First, the seasonal trends of $\delta^2$H were reversed, resulting in a good agreement with
5  the modelled seasonality. Second, measured $\delta^2$H and lc-excess showed consistently lower values as compared to the heather sites, by 5-24 ‰for $\delta^2$H and 4-13 ‰for lc-excess ($\delta^{18}$O was only slightly positively biased, not shown). Aa a consequence, simulations showed a permanent positive offset for Scots pine water use despite consistent seasonality.

    Isotopic variability was comparatively much lower for the groundwater both in time and across monitored wells, and a general agreement was found in the simulations (Fig. 6). The deuterium signal was robustly reproduced, with all measured values falling in the 90% -spread of simulation ensemble. However, the model tended to slightly underestimate lc-excess, with simulated values near zero while measurements were mostly centered on 3 ‰. In addition, the short-term lc-excess variability
was somewhat underestimated in the riparian area.



**Figure 4. (a-c)** Time series of soil volumetric water content ($\theta$) and isotopic composition ($\delta^2$H and lc-excess) at the peat site indicated by purple cross in the bottom maps. Measurements at a given depth are shown in black while colours display the ensemble medians and 90%-intervals of simulations in the two uppermost soil layers. **(d-g)** Model-minus-data difference at two given days when samples were collected in the valley bottom, for deuterium **(d-e)** and lc-excess **(f-g)**; black symbols indicate an absence of sample on one of the two dates.





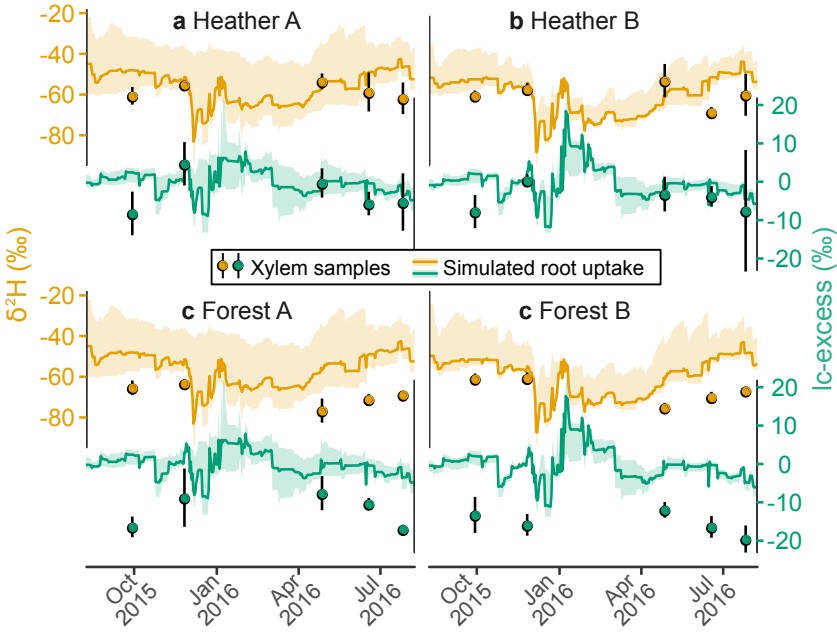

**Figure 5.** Time series of deuterium composition (orange) and lc-excess (green) in the xylem of two heather shrublands (Heather Site A and B) and two Scots pine stands (Forest Site A and B). Measurements are shown with symbols with one standard deviation across replicas, while solid lines display ensemble medians and 90%-intervals of simulations.

Figure 7 shows the model-data comparison at the catchment outlet. The flow (Fig. 7a) and overall signal of stream water $\delta^2$H (Fig 7b) was well reproduced by the model, with consistent "transition" periods of progressive enrichment when atmospheric demand increased and the catchment got drier. Most behavioural models in the ensemble did not completely capture the full extent of winter $\delta^2$H depletion, and the seasonal minimum of $\delta^2$H generally fell below the 90%-spread of the ensemble. However, seasonal variations of modelled lc-excess in the stream were in phase with the datasets throughout the study period: minimum in summer, maximum in winter, although simulated variability was more damped than for $\delta^2$H, with a slight negative bias.

A summary of model performance is shown in Fig. 8 for all sites/compartments, using the dual space of model-data linear correlation (x-axis) and model-to-data ratio of variability (observation-normalized standard deviations, y-axis). Most model-data correlations were significantly positive between 0.5 and 0.85, while insignificant or negative correlations were mostly found where only a few data points were available (xylem) or where seasonal variability was low (e.g. groundwater). In addition, $\delta^2$H variability ratio between model and data was often close to 1, while values for lc-excess were more variable. Some model overestimations of isotopic variability were evident, most often in the topsoil layer (Forest B and Heather sites) but also in the second layers for lc-excess at the Gley site. Interestingly, median model-data agreement in topsoil at Forest site A significantly differed between 2013 (mobile water sampling via lysimeters) and the 2015-2016 period (bulk water sampling via direct equilibration). This was notable in the dramatic increase of model-data correlation and near-one variability ratio for





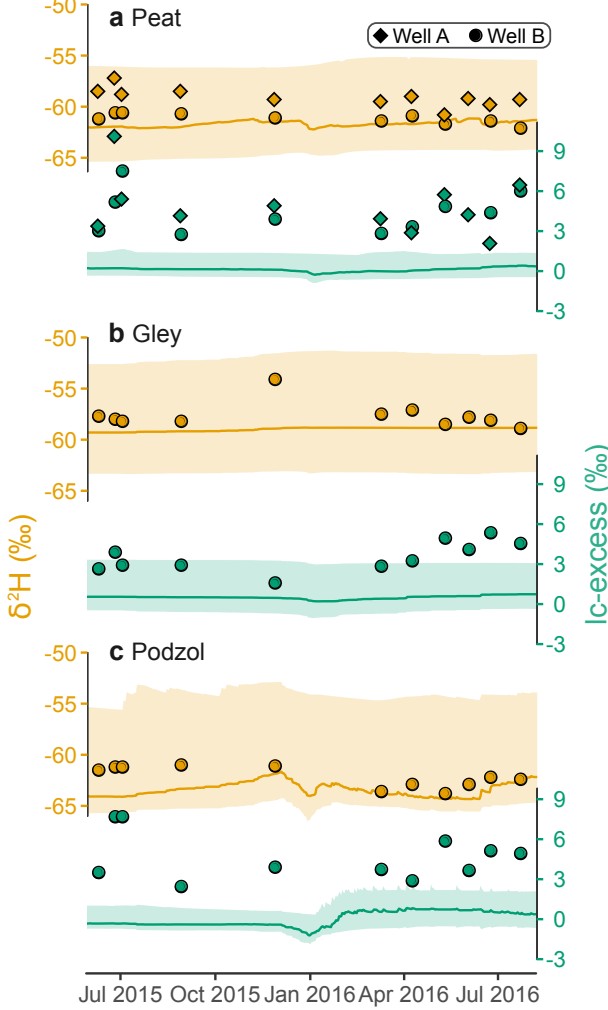

**Figure 6.** Time series of deuterium composition (orange) and lc-excess (green) in groundwater at different locations in the catchment. Measurement are shown with symbols – with two wells on the same simulated peat grid cell, on opposite sides of the stream –, while solid lines and ribbons show the median and 90%-confidence interval of ensemble simulations.

topsoil $\delta^2$H in the latter case, which is consistent with the hypothesis that the simulated soil water composition represents that of bulk water.

## 4.2 Simulated hydrometric and isotopic patterns

Figure 9 provides a spatially-distributed, seasonal view of the ensemble-median of outgoing water fluxes across the catchment over the simulation period. Lateral connectivity was markedly higher during the wetter first half of the hydrological year (October - March, Fig. 9a-b). During this colder, most energy-limited period, surface runoff was significant in many cells where



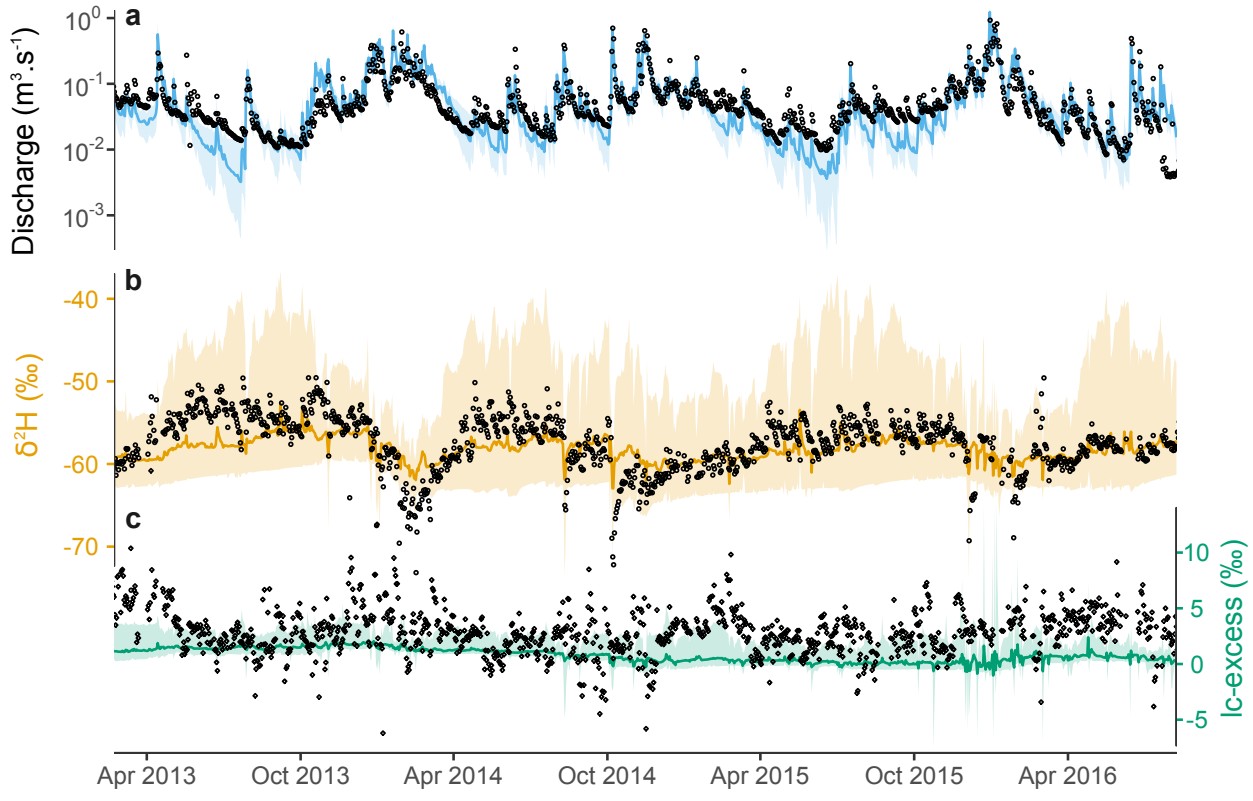

**Figure 7.** Time series of stream **(a)** discharge and isotopic composition – **(b)** $\delta^2$H and **(c)** lc-excess – at the catchment outlet. Measurements are shown with black open symbols while colours display the medians and 90%-confidence intervals of ensemble simulations.

the slopes transition to the valley bottom (up to 53 mm·d$^{-1}$, i.e., 0.006 m$^3$·s$^{-1}$), as well as on some surrounding hillslopes in the southern/south-western part of the catchment (Fig. 9a). Throughout the spring-summer, very few of these overland corridors were usually hydrologically active. In parallel, lateral subsurface connectivity in autumn-winter time was quite widespread across the catchment, particularly concerning the two southernmost stream tributaries where subsurface flux largely exceeded surface runoff (up to 90 mm·d$^{-1}$, Fig. 9b). Some of these subsurface connections were still active during the growing season,

albeit weaker (< 40 mm·d$^{-1}$). Given the predominance of subsurface flow near the channel, return flow dominated the vertical water budget (exfiltration minus infiltration > 0) throughout the year at junctions with the main stream and further downstream, especially in the winter (Fig. 9c). The rest of the catchment was dominated by infiltration, with average net rates of a few mm·d$^{-1}$. Evaporative losses of soil water were much smaller and had a different seasonality than infiltration and throughflow (Fig. 9d-e). In autumn-winter, soil evaporation ($E_s$) was similar in magnitude to ecosystem transpiration ($T_c$, integrated over

all vegetation cover for each grid cell), although at local scales both fluxes remained below a few tenths of mm·d$^{-1}$ (catchment average: 0.11 mm·d$^{-1}$ for both $E_s$ and $T_c$). Conversely, ecosystem transpiration clearly dominated during the rest of hydrological year, with a catchment-averaged rate almost four times higher than that of soil evaporation (0.61 versus 0.16 mm·d$^{-1}$,



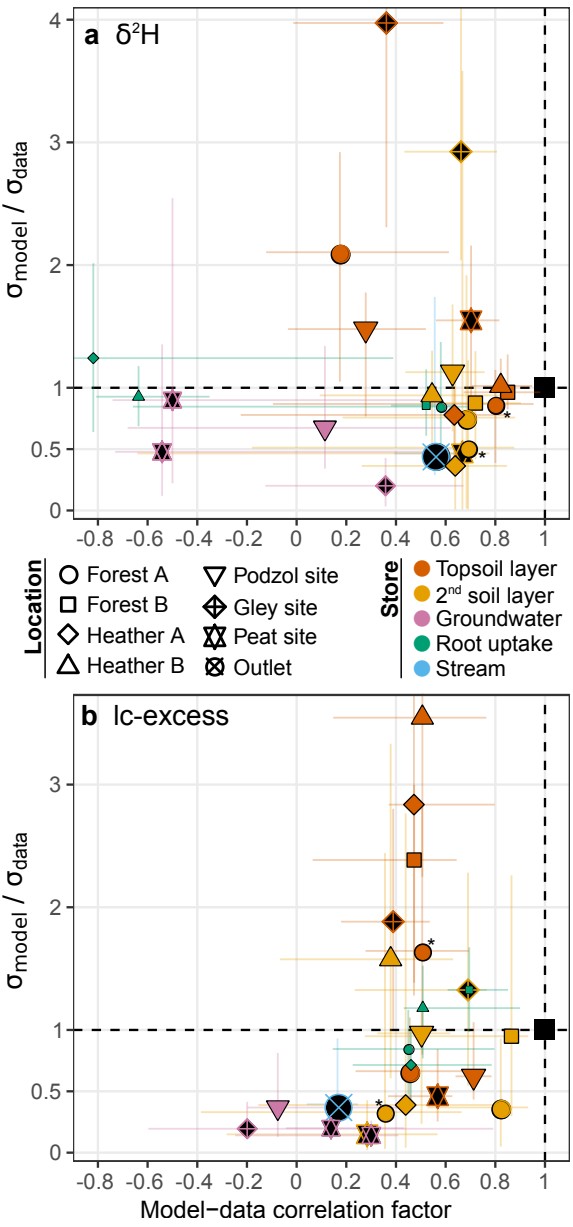

**Figure 8.** Summary of model performance in the dual space of model-data correlation factor and ratio between modelled and observed time series, shown for **(a)** $\delta^2$H and **(b)** lc-excess, showing the median and 90%-spread over the ensemble. The size of each symbol is proportional to the logarithm of the number of daily field data points available. Performances in soil compartments at Forest site A are further separated between periods 2013 and 2015-2016 (the latter indicated with an asterisk), corresponding to two separate field data collection campaigns. Two groundwater wells are presents at the peat site.



respectively). In both cases, the highest values were found in the riparian area, although the spatial contrast was more marked for soil evaporation.

**Figure 9.** Seasonally-averaged daily outgoing water fluxes in the Bruntland Burn over the period 2013 – 2016, showing the ensemble median of simulated **(a-b)** cell-to-cell lateral flow, **(c)** net vertical liquid flow and **(d-e)** evaporative losses.




This spatio-temporal variability in water fluxes was somewhat reflected in that of isotopic patterns (Figs. 10-11). $\delta^2$H in the topsoil went from markedly depleted winter values (average: -61 ‰) to maximum enrichment in spring-summer with larger spatial variability (average: -44 ‰) (Fig. 10a). These temporal variations were well within that of $\delta^2$H in precipitation inputs (Fig. 2c). Yet, the increasing spatial variability of topsoil $\delta^2$H in spring-summer, and the much more pronounced relative seasonality of topsoil lc-excess (Fig. 11a) (compared to that in precipitation, Fig. 2c), indicated a significant influence of evaporation fractionation on isotopic patterns. During the spring-summer period the highest $\delta^2$H values, and most negative lc-excess values, were found in the organic soils of the valley bottom and on the higher hillslopes. These locations are where soil evaporation was highest (Fig. 9d) or where the soils are thinnest (rankers regosols, Fig. 2a). The effect of isotopic fractionation crucially depends on relative storage change (Eq. 7), thus it had large values either because absolute evaporation was high (valley bottom) or because the available storage was limited (thin soils). Conversely, spring-summer lc-excess values were near zero (or even slightly positive), and $\delta^2$H enrichment less pronounced, in most of the topsoil grid cells where the stream is also present, corresponding to the locations where upslope-routed groundwater exfiltrated (Fig. 9c). Finally, positive winter values for lc-excess across the catchment's topsoil hints at a widespread dominance of winter precipitation and mixing processes (via surface connectivity and infiltration, Fig. 9), over fractionating ones.

The isotopic signature in water used by plants for transpiration largely displayed a damped reflection of the topsoil patterns (Figs. 10-11 b). This reflects distributed root uptake across the soil profile, reaching deeper soil compartments where seasonal isotope dynamics were less marked. One consequence is that the model simulated more isotopically-depleted plant water use in the thin regosols of the upper hillslopes compared to the very shallow topsoil layer (northern and western parts of the catchment).

Finally, groundwater $\delta^2$H patterns were comparatively more uniform across the catchment ($\sigma_{spatial}$ = 1.9 ‰) and across seasons (Fig. 10c). Most depleted values were found in the podzolic hillslopes and across the valley bottom, a feature more marked in winter and spring. Lc-excess mostly displayed positive values throughout the year, except for some weakly negative autumn values on the higher hillslopes (Fig. 11c). Markedly positive values were generally found in the organic soil of the valley bottom where fluxes converge. Note that positive values were more spatially homogeneous during winter and spring time, highlighting subsurface recharge lagging behind the more superficial compartments by a few months.

### 4.3 Water ages

Simulated water ages showed significant variability across locations in the catchment, as well as a marked seasonality at most sites selected for the analysis (Fig. 12). For convenience, the sites chosen for analysis in Fig. 12a-d were the same as those where isotopic model-data evaluation was conducted. In general, modelled water age increased with distance downhill, consistent with freely-draining hillslopes sustaining groundwater fluxes into the riparian area. In the soils, water age ranged from a few weeks on the hillslopes to several years in the valley bottom peat where the top soil is affected by exfiltration of older groundwater from upslope areas. Groundwater age was more homogeneous across the watershed but still showed significant differences, averaging one year of age in the podzol-covered locations, compared to 2-to-3 years in the riparian area. Seasonal variations were most significant on the hillslopes, from week-old waters in winter to water ages of 2-to-6 months during





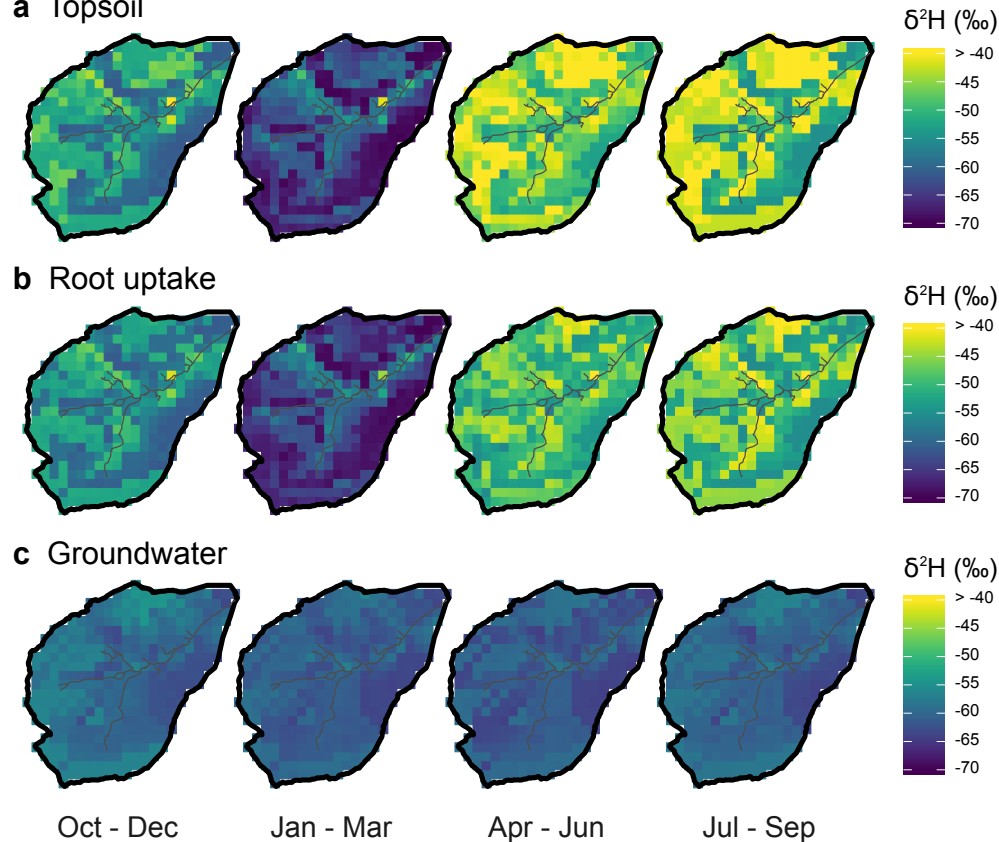

**Figure 10.** Seasonal average of $\delta^2$H in the Bruntland Burn over the period 2013 – 2016, showing the ensemble median of simulations in **(a)** the topsoil layer, **(b)** root water uptake (summed of vegetation covers) and **(c)** groundwater.

the growing season in the vadose zone. Weaker intrinsic seasonal variability was generally found in groundwater, which is consistent with the very flat simulated isotope dynamics (Fig. 6). The age of water uptaken by plants followed the topsoil age patterns in most cases, reflecting the relatively young water ages from shallow rooting depths. One exception is Forest site A,

where the contribution of older water from the second soil layer during the growing season had a clear effect on the age of the water used by vegetation. This latter site interestingly displayed older water ages compared to other hillslopes locations, suggesting slower drainage conditions, likely linked to less marked local topography and receipt of older water from upslope. In addition, the gley site displayed a rather dynamic behaviour in the upper soil layers, similar to podzols, while the turnover of groundwater there was the lowest among all locations, suggesting it is confined and disconnected from the soil profile. Finally,

at the peat site, younger water ages were found in groundwater compared to upper soil layers. This surprising result was likely linked to permanently saturated soils with limited infiltration (disconnection from the surface and overland flow) and recharge (disconnection from confined groundwater) where lateral soil water movement was not simulated, but lateral transfers and mixing occurred in the underlying groundwater.





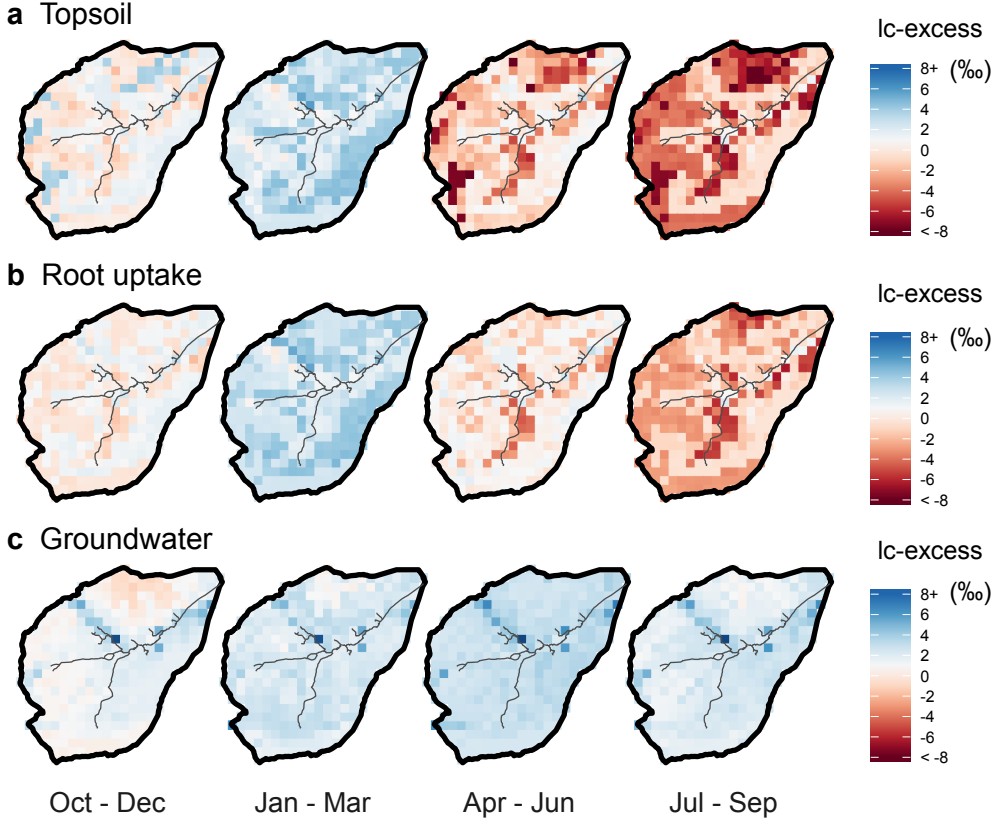

**Figure 11.** Seasonal average of lc-excess in the Bruntland Burn over the period 2013 – 2016, showing the ensemble median of simulations in **(a)** the topsoil layer, **(b)** root water uptake (summed of vegetation covers) and **(c)** groundwater.

Spatial variability was also found in stream water age, as shown in previously referred-to sites and arbitrarily defined loca-

tions along the channel network (Fig. 12e). In two of the main tributaries of the BB (HW2 and HW3), simulated water ages were significantly younger ($\sim$ 0.5-1.5 years) than along the rest of the channel network (1.4$\sim$ 2.8 years). The older water ages found in HW1 might be linked to the presence of high water storage in drift deposits and an extensive raised peat bog in this portion of the valley bottom (Sprenger et al., 2017b), while the streams in HW2 and HW3 emerge further upslope at the drift-free ranker-podzol transition (Fig. 2a). There is a localized increase in water age when moving downstream towards

the peat site, consistent with increased groundwater exfiltration (Fig. 9c) where stream water is a few weeks older than at the catchment outlet. In this lower part of the catchment lower temporal variability was also evident (1.8$\sim$ 2.4 years). Again, this might be derived from groundwater influxes and the extensive presence of saturated peat soils in this part of the catchment, compared to other sections of the stream.





**Figure 12.** Ensemble median of simulated water age at the different sites used for model evaluation (except for Heather site A) in each corresponding compartment (**a-d**), and in the stream at several locations along the channel network (shown on the inset map, **e**) . To improve visibility, all curves have been smoothed using a 7-day moving average window.

## 5 Discussion

### 5.1 Performance of the tracer-enhanced model

The model-data comparison demonstrated that EcH$_2$O-iso captured a significant part of the isotopic behaviour across multiple ecohydrological compartments and landscape positions monitored in the study catchment. Because no calibration was per-





formed on the isotopic components, these results reveal that the water mixing and storage and the water pathways simulated by the hydrologic core of EcH$_2$O-iso correctly reflect the hydrologic dynamics of the basin (Kuppel et al., 2018). Hydrological

states and fluxes in the model evolve driven by local energy gradients (e.g. gravity-driven hydraulic gradient) throughout the landscape, with no direct knowledge about which "water parcels" (e.g., old or young, upslope or downslope) have been mobilized during a given hydrological response (Kirchner, 2003). Conversely, correctly capturing isotope dynamics is conditioned to accurately simulating patterns of water particle velocities, i.e. to routing the correct water parcels all the way from precipitation to their fate in the stream or as evaporative outputs (McDonnell and Beven, 2014). Therefore, the general performances achieved by our celerity-velocity framework give reasonable confidence in the mechanistic description of energy-water-plant couplings adopted by the EcH$_2$O-iso model.

Despite some of the discrepancies presented and discussed below, the overall isotopic model-observation fit is remarkable

because the evaluated ensemble of model configurations was not derived from any tracer-aided calibration, but solely used the information content brought by hydrometric and energy balance datasets in an independent calibration exercise similar to Kuppel et al. (2018). Further, the implementation of water isotope and age tracking, consistent with the original structure of EcH$_2$O and including evaporation fractionation of isotopes, was straightforward and followed well-established methodologies (Eqs. 4-18) without any parameterization specific to the study site. By keeping both the isotopic module and calibration as

minimalistic as possible, our approach avoids adding new, unnecessary degrees of freedom and reduces the risk of overfitting. Specific model performance might thus be lower than what could be achieved using a dual hydrometric-isotope calibration approach (Birkel et al., 2014; van Huijgevoort et al., 2016; Knighton et al., 2017), but because the isotopic signal remains truly independent of the hydrologic calibration our approach allows unique critical analysis and insight into the physical hypotheses underlying simulated flow generation and water mixing.

## 5.2 Insights into critical processes for model future development

The timing of seasonal isotopic dynamics as well as higher-frequency responses were well simulated in the vast majority of cases (summary in Fig. 8), together with value ranges also broadly consistent with observations. Yet, the amplitudes of modelled temporal isotopic responses displayed variable degrees of agreement with that of measured signals. In general, dynamics of deuterium were better reproduced than that of lc-excess, with a trend to underestimate lc-excess in several compartments.

One of these model-data mismatches is the overly enriched signal in the topsoil of the riparian sites during the growing season (Figs. 4, 8). The concomitant underestimated lc-excess hints at an excessive evaporation fractionation signal. As pointed out in section 4.1, these discrepancies can partially derive from the different support represented by model and observations. For instance, the model simulates the composition of the bulk topsoil water, whereas the observations may reflect only the composition of the free draining portion of the soil water. At the long-term riparian locations (Peat and Gley sites, Fig. 2a),

collection by suction lysimeters (Tetzlaff et al., 2014) was used, sampling water under low tension and less affected by fractionation (Brooks et al., 2010; Sprenger et al., 2017b). In addition, the samples from a synoptic field campaign across the extended riparian area in flowing surface waters and ponds on two dates (Lessels et al., 2016) were directly compared to simulated topsoil water (Fig. 4d-g). This was because the current formulation of EcH$_2$O-iso routes all surface water to the next downstream cell





and thus does not account for free-standing water such as the ponds and zero-order streams typically forming outside summer
in the BB, particularly in the north west of the catchment (Lessels et al., 2016). Yet, the sampled surface water in the riparian
area has been shown to have spatially-varying sources, presenting distinctively enriched or depleted $\delta^2$H signals depending if
the source is soil water or groundwater, respectively (Lessels et al., 2016). Systematically comparing soil water to sampled
surface water might thus explain the overestimation of $\delta^2$H, especially in the north-west part of the catchment where limited
groundwater seepage is modelled (Fig. 9c). Secondly, the riparian topsoil in EcH$_2$O-iso might function as an "evaporation
hotspot" to a greater extent than has been found in corresponding sampled surface water locations (Sprenger et al., 2017b).
Indeed, topsoil water is not laterally connected in the model, so that evaporation fractionation remains local (horizontally)
but immediately mixes across the whole layer – as compared to a vertically-stratified isotopic profile in poorly mixed ponded
areas. In addition, while fractionation is modest as compared to other climatic settings at our site, measurements have shown
that ponds and zero-order channels that are not fully evaporated connect to the channel network in spring-summer and drive
the seasonal isotopic enrichment observed in the stream (Sprenger et al., 2017b). Further support for this hypothesis was found
by disabling evaporative fractionation in our simulations: seasonal variability of isotopes in the stream almost completely dis-
appeared, while short-term, event-driven dynamics remained (not shown). Beyond the idiosyncrasies of our study catchment,
and the gap between fine-scale wetland heterogeneity and our model resolution (100×100 m$^2$), a large body of literature has
reported the importance of riparian wetlands as time-varying "chemostats" controlling stream water quality (e.g., Billett and
Cresser, 1992; Smart et al., 2001; Spence and Woo, 2003) or "isostats" mixing isotope signals (Tetzlaff et al., 2014). Since
modelled soil water is not laterally routed to the channel during the onset of the growing season, this might explain some
underestimation of summer $\delta^2$H in the stream outlet, as well as the reported lack of seasonal variability for instream lc-excess
(Fig. 7). Further developments of the model to include ponding effects and/or a more dynamic channel network (rather than
fixed, as currently conceptualised), would help capture these seasonally-varying flow paths in the variably-saturated valley
bottom of low-energy landscapes.

The isotope and age tracking adopts a complete and instantaneous mixing scheme at each sub-time step where water transfers
are computed between the spatially-distributed compartments of the simulated domain. This working hypothesis was chosen
for simplicity, given the wet and cool climate conditions and the relatively long (daily) simulation time steps. The spatio-
temporal variability of simulated fluxes and stores somewhat results in a time-variant partial mixing at the catchment scale
at the stream outlet (van Huijgevoort et al., 2016). However, we note, for example, that our simulations of groundwater lc-
excess showed an underestimated variability and a consistent negative bias towards near-zero values (Fig. 6). It indicates that
the simulated recharge signal is very damped throughout the year and slightly biased towards the signature of over-enriched,
evaporation-affected recharge. This contrasts with the evidenced dominance of winter recharge given the markedly positive
lc-excess values observed at the monitored wells (Scheliga et al., 2017) as well as in other catchments with comparable eco-
climatic settings (O'Driscoll et al., 2005; Yeh et al., 2011; Bertrand et al., 2014). It might point to an exaggerated mixing
across the soil profile in our simulations, overly flattening the precipitation signature and overestimating fractionation signal
in the water percolating to the water table. Given that groundwater directly sustains 19 (±16) % of annual stream flow in our
ensemble simulations (not shown), one can link this lack of variability in groundwater lc-excess to that simulated in stream





water (Fig. 7). While such a link between the degree of unsaturated zone mixing and stream isotopes was not evidenced
by Knighton et al. (2017), there was a much lower contribution of baseflow to discharge in the intermittent catchment they
modelled. More generally, further developments would benefit from incorporating insights from the growing body of literature
on the importance of preferential flow in driving catchment dynamics and tracer mixing (Beven and Germann, 2013). This
would involve implementing conceptualisation of micro-topographic controls on overland flow (Frei et al., 2010) and sub-
surface dual pore space (matrix-macropore) representations of tracer flow paths and mixing (Stumpp et al., 2007; Stumpp and
Maloszewski, 2010; Vogel et al., 2010; Sprenger et al., 2018). Bridging these detailed plot-scale descriptions with a physically-
based ecohydrological model such as EcH$_2$O-iso will likely require a simplified, parsimoniously parameterized implementation
and calibration with tracer data.

   Our modelling experiment also helps to evaluate the conceptualization of isotopic fractionation in the soil water of wet,
energy-limited catchments. The evaporative fractionation is described by the well-established Craig-Gordon model (Craig and
Gordon, 1965), supplemented here with a soil-adapted formulation following Mathieu and Bariac (1996) and Good et al.
(2014). As reviewed by Horita et al. (2008), the Craig-Gordon model is very sensitive to the isotopic composition of atmo-
spheric moisture ($\delta_a$), the relative humidity of the atmosphere at the surface ($h_a$) and the kinetic fractionation factor ($\epsilon_k$).
We assumed isotopic equilibrium between rainfall and atmospheric moisture (Eq. 10), as is commonly done when no direct
measurement of $\delta_a$ is available (Horita et al., 2008). While this empirical, and here spatially-uniform, approach is valid on
monthly time scales in temperate climates (Schoch-Fischer et al., 1983; Jacob and Sonntag, 1991), discrepancies can arise on
shorter time scales and/or when local evaporation significantly feeds atmospheric moisture (Krabbenhoft et al., 1990). Second,
$h_a$ estimates can be a large source of error in wet environments where $h_a$>0.75 (Kumar and Nachiappan, 1999), which is often
the case in our catchment (Wang et al., 2017b). Furthermore, we found a marked sensitivity of isotope dynamics to the strategy
used to calculate $\epsilon_k$ (Eq. 14), consistent with Haese et al. (2013), who found a large impact on simulated soil $\delta^{18}$O in northern
latitudes. We chose to use a formulation based on isotopic diffusivity ratios; the latter were taken from Vogt (1976) because
their experimental protocol covered a comparatively large range of humidity conditions. Yet it seems that very few (if any)
experimental studies estimating these ratios spanned the very humid conditions found at the BB, and further empirical data
could help reduce the associated uncertainties (Horita et al., 2008).

   Finally, we showed that our root uptake simulations for heather shrubs closely matched the measured isotopic signature in
plant xylem. Conversely, a systematic, positive model offset was found for both $\delta^2$H and lc-excess in Scots pines despite the fact
that the model correctly captured the temporal dynamics (Fig. 5). Our simulations assumed identical, exponential root profiles
for all vegetation types within soil types, e.g. the podzol, where these experimental heather and forest sites are found (Kuppel
et al., 2018), thus species-dependent use of soil water from depth-specific isotopic signature cannot be captured. Heather shrubs
have, however, a shallow root system (typically < 5 cm, Geris et al., 2017) and thus its source water might be more affected by
evaporation than Scots pine (which can be deeper-rooted). However, the observed lc-excess values in the soil of Scots pine (-13
to 5.5 ‰; Fig. 3b, Forest site A not shown) were significantly higher than those measured in the pine xylem (-19.6 to -7.6 ‰,
Fig. 5c-d). It mostly seems to stem from significant recorded deuterium depletion while oxygen-18 ratios were consistent or
slightly depleted as compared to soil samples, and we found larger simulations biases (relative to the mean value) in xylem for





deuterium than for oxygen-18 ratios (not shown). Such isotopic departures between soil and xylem water have been reported

in a number of experimental studies, although primarily conducted in seasonally-drier environments (Lin and da SL Sternberg, 1993; Zhao et al., 2016; Vargas et al., 2017). Several mechanisms have been proposed, including a discrimination of heavier isotopes during water uptake controlled by root aquaporins (Mamonov et al., 2007) or mycorrhizal associations (Berry et al., 2017), phloem-xylem water cycling on several time scales (Hölttä et al., 2006; De Schepper and Steppe, 2010; Pfautsch et al., 2015; Stanfield et al., 2017), and stem water evaporation through the bark (Dawson and Ehleringer, 1993). While exploring the relevance of these mechanisms to the ecosystems here simulated goes far beyond the scope if this study, it is clear that the complexity of isotopic dynamics in plant xylem cannot be fully captured simply based on a root-profile-weighted mixing of

soil pools.

## 5.3 Opportunities for characterizing water pathways

The development of EcH$_2$O-iso is a methodological "middle ground" for modelling conservative tracer transport, between detailed plot-scale models across the soil-vegetation-atmosphere continuum (e.g., Mathieu and Bariac, 1996; Melayah et al., 1996; Braud et al., 2005; Haverd and Cuntz, 2010), catchment rainfall-runoff models (Birkel and Soulsby, 2015; McGuire and

McDonnell, 2015; van Huijgevoort et al., 2016; Knighton et al., 2017), and land surface models for earth system studies (Haese et al., 2013; Risi et al., 2016; Wong et al., 2017). This reflects the reasons why the original EcH$_2$O model was developed, namely to provide a physically-based, yet computationally-efficient representation of energy-water-ecosystem couplings where intra-catchment connectivity (both vertical and lateral) could be explicitly resolved (Maneta and Silverman, 2013). The combination of these features is critical, since explicit lateral connectivity (surface, subsurface, and channel) is typically the missing piece in

land surface models (Fan, 2015) and in plot-scale approaches, and the coupling with vegetation processes is typically missing in rainfall-runoff models (van Huijgevoort et al., 2016). The newly developed model provides, for the first time, a transferable, process-based linkage of spatial-temporal patterns of water fluxes (Fig. 9) with those of isotopic tracers (Fig. 10-11) across a headwater catchment.

Here ,a major focus has been put on the isotopic analysis to evaluate the consistency of EcH$_2$O-iso using the wealth of data

available at the study site, and the limitations stemming from the unavoidable technical trade-off we adopted. Yet, principles used for isotope tracking were applied to track water age across the ecohydrological compartments (Fig. 12). This provides a more complete picture of catchment functioning than stream water age, although the latter metric provides an important first-order benchmark for comparison with other modelling approaches. T he mean stream water age of ∼2.1 yrs is consistent with isotope-calibrated rainfall-runoff approaches reporting ∼ 1.55 yrs (van Huijgevoort et al., 2016; Ala-aho et al., 2017)

and ∼1.8 yrs (Soulsby et al., 2015). The low temporal variability found here yields higher discrepancies when considering flow-weighted median ages: ∼2 yrs against 1.2 yrs found by Soulsby et al. (2015) and ∼1 yr reported using transport model driven by StorAge Selection functions (Benettin et al., 2017). We notably find a slower water turnover in the valley bottom soils (∼2.8 yrs) as compared to compared to the spatially-distributed approach of van Huijgevoort et al. (2016) (∼2 yrs), and EcH$_2$O-iso conversely simulates much younger water ages in the groundwater both on the hillslope and in the valley bottom (∼ 0.9 and ∼2.2 yrs, vs. ∼2.9 and ∼3.4 yrs, respectively) and on the hillslope soils (∼0.2 yr vs. ∼0.8 yr) than van Huijgevoort





et al. (2016). Keeping in mind that these discrepancies might arise from differences in modelling and calibration approaches, these mismatches may also confirm a tendency of EcH$_2$O-iso to overemphasize the role of the riparian area as a hydrologic buffer and mixing zone, as well as the contribution of groundwater, in damping the stream isotope response which could be addressed by strategies suggested in preceding sections.

## 6 Conclusions

The EcH$_2$O-iso model presented in this study is, to our knowledge, the first to simulate catchment dynamics of water isotopes ($^2$H and $^{18}$O) and age by combining a physically-based description of hydrological stores and fluxes, a spatially-distributed simulation domain, a predictive vegetation component, and non-conservative isotopic processes (evaporative fractionation). The model has been rigorously tested with an extensive isotopic dataset, in a wide range of ecohydrological compartments of the study region, and has shown very good performance. This indicates that the model is correctly capturing the main

elements of the catchment functioning as seen from both energy celerity and flow velocity viewpoints. Satisfying this dual perspective is key to characterizing water pathways and quantifying the associated travel times in different ecohydrological compartments of headwater landscapes. Complementing more conceptual approaches, the physical basis of the EcH$_2$O-iso model further provides the potential to extrapolate these insights beyond recorded conditions and scales, and to notably project ecohydrological feedbacks of potential environmental changes.

The relatively simple conceptualization of compartment-scale velocities, e.g. assuming complete mixing, and the absence of isotopic calibration, already make the current results particularly encouraging. It also provides a useful framework for hierarchizing model development and benchmarking needs, specific or not to the physics of a high-latitude, low-energy, wet and steep headwater catchment such as the one simulated here. In particular, our results stress the necessary incorporation of partial mixing hypotheses, likely to be critical in drier and/or flatter landscapes where diffusive water movement prevail.

Second, our model-data analysis of isotope dynamics strongly reflects fractionation effects, be it via soil evaporation or species-specific plant water use. Together with the presented model, these considerations will help providing a process-based modelling framework for plot-to-catchment-scale hypothesis testing. This is timely for current challenges in critical zone science, such as exploring the occurrence and mechanisms behind ecohydrological separation of water fluxes (Berry et al., 2017).

*Code and data availability.*  The source code of the EcH$_2$O-iso model is open source (https://bitbucket.org/sylka/ech2o_iso). The datasets

used in this study will be available on the PURE data repository of the University of Aberdeen (https://abdn.pure.elsevier.com).

*Competing interests.*  The authors declare that they have no conflict of interest.





*Acknowledgements.* This work was funded by the European Research Council (project GA 335910 VeWa). M. P. Maneta acknowledges
support from the U.S National Science Foundation (project GSS 1461576) and U.S National Science Foundation EPSCoR Cooperative
Agreement # EPS-1101342. The authors would like to acknowledge the support of the Maxwell compute cluster funded by the University
of Aberdeen. Finally, the authors are grateful to the many people who have been involved in establishing and continuing data collection
at the Bruntland Burn, particularly Christian Birkel, Maria Blumstock, Jon Dick, Josie Geris, Konrad Piegat, Bernhard Scheliga, Matthias
Sprenger, Claire Tunaley, and Hailong Wang, and to Aaron Smith for fruitful discussions regarding the model development.





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
