# Peer review of "EcH2O-iso 1.0: Water isotopes and age tracking in a process-based, distributed ecohydrological model"

_Geoscientific Model Development, 2018_

## Referee Comment (RC1) · Anonymous Referee #1 · 19 Apr 2018

**General comments**

This paper introduces the isotopic tracking of the ecohydrological model EcH2O. The new model development is evaluated using isotope time series from a montane, low-energy catchment in Scotland. The isotope tracking addition to the model is interesting for the GMD readership and the approach is in general well-documented. The availability of isotope time series in different parts of the study catchment is also very useful for gaining scientific insights. However, . . .

1. the paper lacks a clear focus at times, and the writing varies between being very detailed to very general. The authors know the topics very well, and occasionally

make jumps or sweeping descriptions that easily lose the reader. (Examples in specific comments.)

2. Also, the model development rationale is not entirely clear, which makes it difficult to understand whether the evaluation procedure and criteria are sound, well defined, and in proportion to the goals the model are set to achieve.

3. The authors also do not test the sensitivity of neither parameters, mixing assumptions, nor isotope model structure, which limit the insights that could have been generated in the subsequent evaluation process.

4. The authors repeatedly refer to Kuppel et al. (2018) and at times assume the reader to have taken part of it. This is a bit unfortunate, as Kuppel et al. (2018) is not open access (and also was not accessible for me during my review). Please consider including key information, if only in Supplementary information.

5. The paper is lengthy and readability could be improved by e.g., summarising tables and more condensed graphs that can act as reference, or point the reader to the key results (e.g., notations table, definitions table, and scatterplots etc., more figures like Fig 8).

6. The authors mention in their literature review and discussions other models ranging from local to global scale, but it's not clear if the authors mean that their modelling procedure can be scaled up.

7. In the abstract and conclusions, the authors claim that the framework is useful beyond the type of low energy catchment simulated here. However, I feel this statement is misleading and goes well beyond the evidence provided in the paper, and would require e.g. validation in other types of catchments.

8. Equations: subscripts and superscript should be in upright font when constituting a describing word (e.g., out, in, snow etc.) and only in cursive for variables (e.g.,

t). Function names such as "max" and "min" should also be in upright font.

**Specific comments**
Abstract: Very sweeping and general, and raises many questions. Please consider to be more specific. E.g., what is meant by "good [. . .] match in most cases", "powerful tool", "some model development"? What kind of cases, why is it powerful, what kind of model development? And what is the model development rationale? What can the model be used for? "Celerity" – a term used in the abstract, introduction, discussion and conclusion, but not clearly explained in the analyses and results sections.

Introduction: It could be useful for the authors to explain how such their study is linked to practical and societal meaningful issues. For example, the authors explains many times how isotopic characterisation could "provide insights into water pathways", linked to "water flux partitioning", and understanding "catchment functioning", but the reader is left to figure out on her own if these topics are interesting and important also in a broader context. E.g., could improving our understanding of catchment functioning also for example be directly linked to our capacity to design models capable of forecasting floods, and works well under a rapidly changing climate? No need to be lengthy, but just to provide a context. Some interesting debates about evaporation partitioning is also not included, among others: (Coenders-Gerrits et al., 2014; Evaristo et al., 2015; Jasechko et al., 2013; Schlesinger and Jasechko, 2014; Wei et al., 2017).

P2L10: What do the authors mean when writing that the "hydrology has remained simplistic" in land surface models? Please specify. And why are dynamic vegetation models and global hydrological models not mentioned?

P3L9: "evaporative losses in ET". Please consider "terrestrial evaporation".

[Figure]

P3L11: "transpiration (T)". Please consider using "$E_t$" for transpiration, to avoid the confusion with temperature T.

P3L19: Key features are described, but the rationale is not explained. E.g., why the model developed is the described way? What are the authors hoping to achieve?

P3L28: Please consider new paragraph for the research questions.

P3L30: The research questions could be formulated in a more specifically way. E.g., What are "physics"? Are "mixing assumptions" really investigated in this paper? What kind of "implications and opportunities" do the authors have in mind?

P4 Sect 2.1: Please describe the key features and main limitations of the EcH2O model. Including examples of where and for what kind of purposes the model has been used would also be useful.

P5 Fig 1: Please consider illustrating the isotope tracking assumptions within the model chart, e.g., transpiration is not considered fractioning, throughfall is not aging etc.

P6L9: "One exception. . ." Perhaps new paragraph?

P6L14 "No spill-over". Not sure what is meant. There is throughfall, right?

P8L13 "PET" Please consider using $E_{\text{pot}}$, as PET could also be precipitation, evaporation, and temperature.

P11L3-4 "Autumn" Lowercase letters

P13L26-27 "model-to-data ratio of standard deviation and model-data Pearson's correlation factor". Please consider discussion the merits and pitfalls of using these evaluation metrics. See for example (Biondi et al., 2012) for review of different validation procedures that might be of relevance. '

P14 Sect 4.1. The time series section is detailed and provide considerable amount of information. However, it is also difficult for the reader to quickly get a grasp of the main strength and weaknesses of the model. Please consider including e.g., scatterplots.

P21 Fig 9, P23 Fig 10: Possibly consider moving some of the maps to the SI, and condense the information by grouping (by e.g., riparian/upstream/downstream etc types of regions).

P26L9- "By keeping the..." Parts of this could also be modelling rationale that could been useful in the introduction section or model set-up.

P3019: "ecohydrological feedbacks". A bit general, and not clear what the authors mean. Ecosystem response in terms of $CO_2$ fertilisation and root depth development?

**References**

Biondi, D., Freni, G., Iacobellis, V., Mascaro, G. and Montanari, A.: Validation of hydrological models: Conceptual basis, methodological approaches and a proposal for a code of practice, Phys. Chem. Earth, Parts A/B/C, 42–44, 70–76, doi:10.1016/j.pce.2011.07.037, 2012.

Coenders-Gerrits, A. M. J., van der Ent, R. J., Bogaard, T. A., Wang-Erlandsson, L., Hrachowitz, M. and

Savenije, H. H. G.: Uncertainties in transpiration estimates, Nature, 506(7478), E1–E2, doi:10.1038/nature12925, 2014.

Evaristo, J., Jasechko, S. and McDonnell, J. J.: Global separation of plant transpiration from groundwater and streamflow, Nature, 525(7567), 91–94, doi:10.1038/nature14983, 2015.

Jasechko, S., Sharp, Z. D., Gibson, J. J., Birks, S. J., Yi, Y. and Fawcett, P. J.: Terrestrial water fluxes dominated by transpiration, Nature, 496(7445), 347–50, doi:10.1038/nature11983, 2013.

Schlesinger, W. H. and Jasechko, S.: Transpiration in the global water cycle, Agric. For. Meteorol., 189–190, 115–117, doi:10.1016/j.agrformet.2014.01.011, 2014.

Wei, Z., Yoshimura, K., Wang, L., Miralles, D. G., Jasechko, S. and Lee, X.: Revisiting the contribution of transpiration to global terrestrial evapotranspiration, Geophys. Res. Lett., 44(6), 2792–2801, doi:10.1002/2016GL072235, 2017.

---

## Referee Comment (RC2) · Anonymous Referee #2 · 19 Apr 2018

Kuppel et al. presents a physically-based ecohydrological model EcH2O-iso that can track water isotopic tracers (2H and 18O) and age. The EcH2O-iso is an extension of the EcH2O model (Maneta and Silverman, 2013). The EcH2O-iso model was evaluated at the Bruntland Burn catchment in the Scottish Highlands, and the simulation results show reasonable agreements with the isotopic measurements.

The paper is well written and structured, and it could be a potentially useful contribution to the literature. However, the authors used very general terms in many parts of their model evaluation, which makes it difficult to assess the reliability of their results. For example, no statistics were shown on any of the time series plots, so there is no way

that the readers can examine the model performance. Therefore, a major revision is suggested to improve the presentation of the current manuscript.

Specific comments:

Pg2, L9-12: The statement provided here seems not directly related to the paragraph above and below it. It is not clear what was the authors' attempt to deliver here. Also, what is "simplistic" meant by the authors with regard to the hydrology in land surface models?

Pg3, L28-31: It is not clear how these questions being addressed in the paper. It would be very helpful if the authors could add more details about the experimental design to illustrate how these questions were linked to the results.

Pg4, L4: It might be better to change "climate" to "microclimate" since the spatial and temporal scales used in the model is relatively small than the scales used in climate science.

Pg4, L11: What is the temperature threshold for the partitioning between liquid and snow components? How does the model quantify snowpack depth for a given amount of precipitating snow?

Pg4, L12: Canopy conductance is a key factor determining the amount of canopy transpiration. How is canopy conductance represented in the model? Is it simulated at each model time step?

Pg6, L3: $\Delta t$ is redundant here as it has been defined right above eqn (2).

Pg8, L13: Could the authors provide any reference for the amount of PET estimated at the study site?

Pg11, L27-29: Were there any missing data during the measurement period? If so, what was the gap-filling treatment for the meteorological observations? Also, what was the temporal resolution of the meteorological observations?

Pg 12, L 13: How did the authors determine the transient dynamics has been removed after a 3-year spin up period?

Pg 12, L21: Why did the authors set the depth of the first soil layer to 0.001 m? How sensitive does the model respond to the changes in the depth of the first soil layer?

Pg 13, L11: It should be Eq. 20 instead of Eq. 19.

Pg 19, L 26-27: How is the seasonal change of vegetation represented in the model? Was the increase of ecosystem transpiration resulted from the increase of vegetation leaf area or the increase of canopy conductance? Did the authors check the water loss from canopy evaporation? How much of difference did the model simulate between canopy evaporation and soil evaporation?

Pg29, L23: Please change "T he" to The.

Pg30, L13-14: This is a very general statement. It would be very helpful if the authors could revise it with more specific terms so the readers can catch up easily.

Pg30, L14-15: Again, it is difficult for the readers to understand why this wound indicate the model is correct in both energy celerity and flow velocity viewpoints. It might be useful to explain what exactly are the energy celerity and flow velocity viewpoints meant by the authors.

––––––––––––––––––––––––––––

---

## Referee Comment (RC3) · Anonymous Referee #3 · 23 Apr 2018

This review report is for the manuscript, entitled: "EcH2O-iso 1.0: Water isotopes and age tracking in a process-based distributed ecohydrological model" by Kuppel et al.. This study embedded the water isotopic tracers and age into an ecohydrological model, EcH2O and then applied this model onto a small catchment. This model, therefore, could simulate the spatio-temporal variation of water flux and water isotopic composition in soil moisture, plant xylem, and groundwater. Overall speaking, I enjoyed reading this study which, indeed, is a great and innovative work. The spatio-temporal patterns of water isotopes can be demonstrated now and the hypothesis we have been concerned can be tested. The simulation is promising, which indicates that the present concepts and knowledge are tentatively correct. However, there are still

some concerns that should be addressed for completing the statements. First of all, this study simulated the hydrological processes without parameterization and calibration. Although the lack of calibration is a good way to test hypothesis comprehensively, it would lower the practical applicability for transferring this model to other catchments. This Aberdeen catchment with intensive observations is quite unique around the world. Therefore, it would be great to discuss the potential parameterization, particularly for the soil moisture, transpiration, and groundwater. The parameterization could not only increase the applicability for other catchments, but also help to introduce the landscape characteristics into the parameters, which is an important concern of critical zones where researchers attempt to incorporate the geophysical characterization into substance transport. Secondly, the water isotopic measurement in soil moisture is very difficult and tricky. As mentioned by Orlowski et al. (2016), it is intricate to determine the soil water isotopic composition. Presently, this model integrated all soil layers into one storage, which is acceptable, but can the authors explain more on what kind of soil water they simulated and what is their opinion about this issue in modeling work? Thirdly, the simulated and observed deuterium composition and lc-excess in forest sites exist large discrepancies. It was straightforwardly attributed to the dependency among species. It indicated that vegetation pumping has great differences among species (e.g. heather and forest). It will be great if the authors can give some suggestions for further parameterization. Finally, the observed lc-excess values of groundwater are higher than simulated ones indicating the exaggerated mixing across the soil profile. However, evaporation from shallow groundwater could raise the lc-excess variability as well. Can the authors explain more to this concern and provide some thinking for further modeling development?

---

## Author Comment (AC1) · 4 Jun 2018

The authors would like to thank the Referee #1 for her/his valuable comments and suggestions to improve the manuscript. They have been taken into account in the revised manuscript, as follows (original referee's comments in bold):

[Figure]

**General comments**

This paper introduces the isotopic tracking of the ecohydrological model $ECH_2O$. The new model development is evaluated using isotope time series from a montane, low-energy catchment in Scotland. The isotope tracking addition to the model is interesting for the GMD readership and the approach is in general well-documented. The availability of isotope time series in different parts of the study catchment is also very useful for gaining scientific insights. However, . . .

**1. The paper lacks a clear focus at times, and the writing varies between being very detailed to very general. The authors know the topics very well, and occasionally make jumps or sweeping descriptions that easily lose the reader. (Examples in specific Comments.)**

We thank the referee for this comment. In general, we have tried to make the narrative more consistent as specified in the corresponding specific comments below.

**2. Also, the model development rationale is not entirely clear, which makes it difficult to understand whether the evaluation procedure and criteria are sound, well defined, and in proportion to the goals the model are set to achieve.**

The objective of the paper is to describe and demonstrate the development of a flux/age tracking component built on an existing ecohydrological model. The rationale for this new development is twofold. First, the tracking component is added to a spatially-distributed energy and water balance model with a strong physical base that explicitly simulates the spatio-temporal heterogeneity of the water mixing processes. Then, we evaluate how this ecohydrological model calibrated solely on hydrometric/energy balance data could simulate spatio-temporal isotope variations without any additional calibration of the tracking and fractionation components. Because of the diversity of fluxes and storage dynamics tracked in the model, we put the emphasis on testing the new model with a wide range of isotopic datasets, and use visual inspection and generic

quantitative metrics (such as mean absolute error and model-data correlation, see response to the corresponding specific comment) for a generic evaluation and further discussion. In the future, when moving towards more operational purposes, specific calibrations of the isotopic component using metrics such as KGE or NSE may be beneficial. Note that specific aspects of this discussion relevant to the model development rationale, and the evaluation metrics are further addressed below in specific Comments.

**3. The authors also do not test the sensitivity of neither parameters, mixing assumptions, nor isotope model structure, which limit the insights that could have been generated in the subsequent evaluation process.**

A comprehensive sensitivity analysis of parameters was already performed by Kuppel et al. (2018), along with a description of the ensemble of parameters used in this paper, which were derived from a multi-objective calibration method conducted using constraints from hydrometric and energy balance observations (see Sect. 3.3). Additional parameter sensitivity analysis and calibration using isotopes datasets would provide complementary information to further constrain parameter uncertainty. However, by doing so we would lose an opportunity to assess how the original ECH$_2$O structure performs against a dataset that is truly independent form the standard hydrometric information typically used in model calibration/validation exercises. A comparison of the performance of different mixing models is beyond the scope of the paper. The presented model simulates isotope tracking using a simple full mixing assumption, which avoids hard-to-test partial or incomplete mixing hypotheses and therefore permits an in-depth discussion of model strengths and weaknesses for potential applications and hypothesis-driven model developments.

**4. The authors repeatedly refer to Kuppel et al. (2018) and at times assume the reader to have taken part of it. This is a bit unfortunate, as Kuppel et al. (2018) is not open access (and also was not accessible for me during my review). Please consider including key information, if only in Supplementary information.**

It is difficult not to refer extensively to Kuppel et al. (2018) because it gives key details such as a description of improvements to the original ECH$_2$O model, a quantification of the model performance on the study site and also describes the basic configuration used to evaluate ECH$_2$O-iso. While it is not possible to reproduce all this information in the paper, we strived to include the information relevant for the interpretation of the results of the present study. Nevertheless, we recognize this can be frustrating. To ameliorate this problem we revised the manuscript to add further details regarding model development rationale, key features and limitations, and the range of environments on which the model has been successfully applied. We have also added a figure to the Supplementary Information that indicates the time spans used for calibration and evaluation (Fig. S1) and the list and description of the calibrated parameters (Table S1, adapted from Kuppel et al., 2018):

*"**Table S1.** Calibrated parameters used in this study, grouped according to their four components: soil units or vegetation types."*

[Figure]

| Name | Description |
|------|-------------|
| **Soil-distributed** (Peat, Gley, Podzol, Ranker) | |
| $D_{soil}$ | Total soil depth (m) |
| $D_{L1}$ | Depth of the 1st hydrological layer (m) |
| $D_{L2}$ | Depth of the 2nd hydrological layer (m) |
| $\phi$ | Porosity ($m^3.m^{-3}$) |
| $K_{hx}$ | Saturated horizontal hydraulic conductivity ($m.s^{-1}$) |
| $K_{hratio}$ | Ratio of vertical-to-horizontal hydraulic conductivity (–) |
| $\lambda_{BC}$ | Brooks-Corey exponent parameter (–) |
| $\Psi_{ae}$ | Air-entry pressure head (m) |
| $\theta_r$ | Residual soil moisture ($m^3.m^{-3}$) |
| $k_{root}$ | Exponential root profile ($m^{-1}$) |
| **Vegetation-distributed** (Pine, Hather, Moss Grass) | |
| $gs_{max}$ | Maximal stomatal conductance (m.s-1) |
| $CWS_{max}$ | Maximum interception storage per unit LAI (m) |
| $T_{opt}$ | Optimal photosynthesis temperature (C) |
| $\Psi_d$ | Soil water potential halving stomatal conductance (-m) |
| $c$ | Sensitivity of stomatal conductance to soil water potential (–) |
| $K_{beer}$ | Light attenuation coefficient (–) |

(see Figure at the end of the response) *"**Figure S1.** Temporal windows –at daily resolution– covered by each of the datasets (orange) at the different sites (italic font) grouped by observation type (bold font) used to calibrate the ECH$_2$O-iso model, while the full simulation period (blue) is used for evaluating the isotopes and age tracking module."*

**5. The paper is lengthy and readability could be improved by e.g., summarising tables and more condensed graphs that can act as reference, or point the reader to the key results (e.g., notations table, definitions table, and scatterplots etc., more figures like Fig 8).**

Striking a balance between providing sufficient detail while keeping the paper concise is challenging. We moved away much of the methodological details to Supplementary Materials or to Kuppel et al (2018) and much of the bulk of the paper describes and discusses results on the temporal and spatial patterns of water compositions and age, which are key foci of our study. Nonetheless, we have edited the manuscript to reduce verbosity and have moved the section and figures on lc-excess (Fig. 11) to Supplementary Information, as they offered similar information to Fig. 10.

**6. The authors mention in their literature review and discussions other models ranging from local to global scale, but it's not clear if the authors mean that their modelling procedure can be scaled up.**

The review was meant to contextualize the model within the state of the art, and to indicate that other similar models with different strengths and weaknesses exist. However, one of the features of the $ECH_2O$ model is that it can be run at a wide range of spatial scales, provided that the necessary inputs are available. Indeed, its spatial domain is constructed and determined by a regular-gridded digital elevation model (DEM) map that defines the topography and the drainage network, and establishes the finite-differences grid on which the governing equations are solved (Maneta Silverman, 2013). Currently applications have been conducted at the plot scale (Maneta Silverman, 2013; Douinot et al., *Plot scale modelling to asses forest effects on water partitioning and flux ages*, in prep.), in small catchments (1-10 km2) (Kuppel et al., 2018; Lozano-Parra et al., 2014), in larger watersheds and small regions ($10^2$-$10^3$ km$^2$) (Maneta and Silverman, 2013; Simeone, 2018). While these studies obviously did not include isotopic tracking, the hydrologic core is the same. In the revised manuscript, we added the following sentence in the model description (sect. 2.1) in order to emphasize this multi-scaling potential:

*"[. . .] relative humidity, and wind speed). In addition, the flexible definition of the spatial domain in $ECH_2O$ allows for applications at a range of scales: from the plot (Maneta and Silverman, 2013), to small catchments (1-10 km$^2$ – Lozano-Parra et al., 2014;*

*Kuppel et al., 2018), to larger watersheds ($10^2$-$10^3$ $km^2$ – Maneta and Silverman, 2013; Simeone, 2018)."*

**7. In the abstract and conclusions, the authors claim that the framework is useful beyond the type of low energy catchment simulated here. However, I feel this statement is misleading and goes well beyond the evidence provided in the paper, and would require e.g. validation in other types of catchments.**

The abstract has been edited to avoid making such claim (see the first specific Comment below). In the conclusions however, we argue that the ECH$_2$O-iso was not specifically designed for simulating the kind of catchment here studied and that it has applicability to other regions. In addition, at present the implementation of isotope and age tracking also avoids any location-specific parameterization. From a methodological viewpoint, the specificity of our site lies not so much in its environmental conditions but rather in the richness of available datasets. As a result, there is no reason to think that our methodology (including the ECH$_2$O-iso model) could not perform well in other environments. The conclusions have been modified in the revised manuscript to emphasize this aspect (P30L14):

*"Despite some limitations, this isotope-based evaluation suggests a reasonable capture of the velocity fields (i.e., how fast water parcels move) across the catchment, and complements a previous calibration and evaluation mostly using hydrometric observations (water fluxes and storage dynamics) which indicated a good simulation of catchment functioning from a celerity viewpoint (i.e., how fast energy propagates via the hydraulic gradient) (Kuppel et al., 2018). Satisfying this dual velocity-celerity perspective is key to characterising water pathways and quantifying the associated travel times in different ecohydrological compartments of headwater landscapes. Complementing more conceptual approaches, the physical basis of the ECH$_2$O-iso model further provides the potential to extrapolate these insights beyond recorded conditions and scales, and to notably project the reciprocal feedbacks between plant water use, hydrological pathways and potential environmental changes. The relatively simple con-*

*ceptualisation of compartment-scale velocities, e.g. assuming complete mixing and without site-specific parameterization, and the absence of isotopic calibration, already make the current results particularly encouraging. It also provides a useful framework for hierarchising model development and benchmarking needs. For example, some of the model-data discrepancies in our results stress the necessary incorporation of partial mixing hypotheses, likely to be critical in drier and/or flatter landscapes where diffusive water movement prevails. Second, our model-data analysis of isotope dynamics strongly reflects fractionation effects, be it via soil evaporation or species-specific plant water use. Finally, the versatility of climatic settings in which the original ECH$_2$O model has already been evaluated facilitates applying the presented methodology beyond the specifics of a high-latitude, low-energy, wet and steep headwater catchment such as the one simulated here. Further, the flexible spatial domain used by the model will help providing a process-based modelling framework for plot-to-catchment-scale hypothesis testing. This is timely for current challenges in critical zone science, such as exploring the occurrence and mechanisms behind the postulated ecohydrological separation of water fluxes (Berry et al., 2017)."*

**8. Equations: subscripts and superscript should be in upright font when constituting a describing word (e.g., out, in, snow etc.) and only in cursive for variables (e.g., t). Function names such as "max" and "min" should also be in upright font.**

We thank the referee for this suggestion. All subscript and superscript notations, as well as function names, have been formatted accordingly in the revised manuscript.

**Specific Comments**

**Abstract: Very sweeping and general, and raises many questions. Please consider to be more specific. E.g., what is meant by "good [. . .] match in most cases", "powerful tool", "some model development"? What kind of cases, why**

[Figure]

**is it powerful, what kind of model development? And what is the model development rationale? What can the model be used for? "Celerity" – a term used in the abstract, introduction, discussion and conclusion, but not clearly explained in the analyses and results sections.**

We thank the Referee for this. We have edited the abstract to add precision, as well as to add specificity to the rationale and potential model applications. We also addressed the issue of making "celerity" easier to understand / redefined when used here and in other parts of the manuscripts (see other corresponding comments). In the revised manuscript, the abstract now reads as follows:

*"We introduce $ECH_2O$-iso, a new development of the physically-based, fully-distributed ecohydrological model $ECH_2O$ where the tracking of water isotopic tracers ($^2H$ and $^{18}O$) and age has been incorporated. $ECH_2O$-iso is evaluated at a montane, low-energy experimental catchment in northern Scotland using 16 independent isotope time series from various landscape positions and compartments; encompassing soil water, groundwater, stream water, and plant xylem. The results show consistent isotopic ranges and temporal variability (seasonal and higher-frequency) in across the soil profile at most sites (especially on hillslopes), a broad model-data agreement in heather xylems, and consistent deuterium dynamics in stream water and in groundwater. Since $ECH_2O$-iso was calibrated only using hydrometric and energy flux datasets, tracking water composition provides a truly independent validation of the physical basis of the model for successfully capturing catchment hydrological functioning, both in terms of celerity of energy propagation shaping the hydrological response (e.g. runoff generation under prevailing hydraulic gradients), and of flow velocities of water molecules (e.g., in consistent tracer concentrations at given locations and times). Additionally, we also show that the spatially-distributed formulation of $ECH_2O$-iso provides the possibility to quantitatively link water stores and fluxes with spatio-temporal patterns of isotopes ratios and water ages. However, our study case also highlights model-data discrepancies in some compartments, such as an over-dampened variabil-*

*ity in groundwater and stream water lc-excess, and over-fractionated riparian topsoils. The adopted minimalistic framework, without site-specific parameterization of isotopes and age tracking, facilitates the interpretation of these mismatches into model development and benchmarking needs, while taking into account the idiosyncracies of our study catchment. Notably, we suggest that more advanced conceptualisation of soil water mixing and of plant water use would be needed to reproduce some of the observed patterns. Balancing the need for basic hypothesis testing with that of improved simulations of catchment dynamics for a range of applications (e.g., plant water use under changing environmental conditions, water quality issues, and calibration-derived estimates of landscape characteristics), further works could also benfit from including isotope-based calibration."*

**Introduction: It could be useful for the authors to explain how such their study is linked to practical and societal meaningful issues. For example, the authors explains many times how isotopic characterisation could "provide insights into water pathways", linked to "water flux partitioning", and understanding "catchment functioning", but the reader is left to figure out on her own if these topics are interesting and important also in a broader context. E.g., could improving our understanding of catchment functioning also for example be directly linked to our capacity to design models capable of forecasting floods, and works well under a rapidly changing climate? No need to be lengthy, but just to provide a context. Some interesting debates about evaporation partitioning is also not included, among others: (Coenders-Gerrits et al., 2014; Evaristo et al., 2015; Jasechko et al., 2013; Schlesinger and Jasechko, 2014; Wei et al., 2017).**

We thank the Referee for bringing this perspective. In the revised manuscript, the end of the first paragraph of the Introduction has been modified in this regard (P2L6):

*"[...] water pathways at scales ranging from the pedon (Sprenger et al., 2018) to the catchment landscape (McGuire and McDonnell, 2006; Birkel and Soulsby, 2015). At larger scales, such approaches can yield global estimates of terrestrial water flux par-*

[Figure]

*titioning (Good et al., 2015), where recent scrutiny has been brought upon separating plant transpiration from other source of evaporative losses (e.g., Jasechko et al., 2013; Coenders-Gerrits et al., 2014; Schlesinger and Jasechko, 2014; Wei et al., 2017). At catchment and watershed scales, an understanding of landscape functioning in turn helps designing robust models to predicts the impact of climate extremes and environmental changes in society-relevant issues such as water resources management, flood forecasting, and impact assessment of land cover – land use change (e.g., Troy et al., 2015; Zhang et al., 2017)."*

**P2L10: What do the authors mean when writing that the "hydrology has remained simplistic" in land surface models? Please specify. And why are dynamic vegetation models and global hydrological models not mentioned?**

Given the scope of the paper, we only refer to land surface models on which (to our knowledge) isotopes tracking have been implemented: JSBACH (Haese et al., 2013), ORCHIDEE (Risi et al., 2016) and CLM (Wong et al., 2017). These land surface model have simplified descriptions of the hydrologic system that do not include explicit laterals water transfers (or is represented using a calibrated residence time for linear storage decrease), shallow and deeper subsurface flows, or channel routing. This is made clearer in the revised manuscript. Some of these models also include vegetation dynamics, and this will be mentioned. To our knowledge, no global hydrological model integrates isotope tracking. The revised manuscript has been modified as follows:

*"[. . .] While the simulation of energy budgets and biogeochemical cycles is increasingly detailed in these land surface models -sometimes including vegetation dynamics- the hydrology has, however, remained somewhat simplistic (or even absent) regarding lateral transfers as overland flow, shallow and deeper subsurface flows and channel routing (Fan, 2015). This makes it difficult to take advantage of isotopes tracking to characterise the role of cascading downstream water redistribution in the spatial patterns of catchment functioning. [. . .]"*

[Figure]

**P3L9: "evaporative losses in ET". Please consider "terrestrial evaporation".**

It has been modified to "terrestrial evaporation" in the revised manuscript. In addition, we now use "E" instead of "ET" to refer to evapotranspiration.

**P3L11: "transpiration (T)". Please consider using "Et " for transpiration, to avoid the confusion with temperature T.**

"T" has now been replaced by "$E_t$" throughout the revised manuscript.

**P3L19: Key features are described, but the rationale is not explained. E.g., why the model developed is the described way? What are the authors hoping to achieve?**

We have modified the introduction so that the rationale of the original $ECH_2O$ development explains our choice for developing an isotopes and age tracking module (P3L20): *" Here, we implement isotope and age tracking in the physically-based, fully-distributed model $ECH_2O$ (Maneta and Silverman, 2013). This model was chosen because of it provides a physically-based, yet computationally-efficient representation of energy-water-ecosystem couplings where intra-catchment connectivity (both vertical and lateral) can be explicitly resolved. In addition, $ECH_2O$ separately solves the energy balance at the top of the canopy and at the soil surface, allowing a process-based separation of Es, Et, and Ec. The novel isotopic and age [. . .]"*

**P3L28: Please consider new paragraph for the research questions.**

**P3L30: The research questions could be formulated in a more specifically way. E.g., What are "physics"? Are "mixing assumptions" really investigated in this paper? What kind of "implications and opportunities" do the authors have in mind?**

We answer to these two comments jointly. These are good suggestions, and the research questions are now shown as a list in the revised manuscript, for further clarity. In addition, these questions have been modified as follows: *"We ask the following*
*questions:*

- *To what extent can a hydrometrically-calibrated, physically-based hydrologic model correctly reproduce internal catchment dynamics of isotopes?*

- *What are the limitations of these isotopic simulations? Do they relate to the underlying model physics and/or to the tracking approach adopted?*

- *How useful and transferrable is this model framework for simulating spatio-temporal patterns of isotopes and water ages?"*

**P4 Sect 2.1: Please describe the key features and main limitations of the ECH$_2$O model. Including examples of where and for what kind of purposes the model has been used would also be useful.**

We have extended the first paragraph of Sect. 2.1 to include a further description of ECH$_2$O and examples of past applications:

*"[. . .] relative humidity, and wind speed). In addition, the flexible definition of the spatial domain in ECH$_2$O allows for applications at a range of scales: from the plot (Maneta and Silverman, 2013), to small catchments (1-10 km$^2$ – Lozano-Parra et al., 2014; Kuppel et al., 2018), to larger watersheds ($10^2$-$10^3$ km$^2$ – Maneta and Silverman, 2013; Simeone, 2018). Despite some potential limitations due to the absence of diffusion-driven water redistribution or an explicit biogeochemical cycle providing ecosystem respiration, to date the model yielded satisfactory results and insights across the diversity of climatic settings (semiarid to humid/energy-limited) and scientific focuses (e.g., water balance, energy balance, or plant hydraulics) covered by the aforementioned studies. A comprehensive description of ECH$_2$O can be found [. . .]"*

**P5 Fig 1: Please consider illustrating the isotope tracking assumptions within the model chart, e.g., transpiration is not considered fractioning, throughfall is not aging etc.**

This a good suggestion, we have modified Fig. 1 and its caption, so that fractionating processes appears more explicitly but we kept the throughfall assumptions in the main text in order not overload the figure (the revised figure can be found at the end of this document):

*"**Figure 1.** Water compartments (black rectangles) and fluxes (coloured arrows) as represented in ECH$_2$O, with the dashed arrows indicating processes where isotopic fractionation is simulated. The numbers between brackets reflect the sequence of calculation within a time step. Note that water routing (steps [8] to [13]) differs between cells where a stream is present (∘) or not (∗)."*

**P6L9: "One exception. . ." Perhaps new paragraph?**

It has been amended in the revised manuscript.

**P6L14 "No spill-over". Not sure what is meant. There is throughfall, right?**

By "no spill-over", we meant that since in the ECH$_2$O model "canopy drainage occurs at the rate at which precipitation increases above the maximum canopy storage" (Maneta and Silverman, 2013), and because maximum canopy storage is constant in our simulations, only the precipitation from the current time step can contribute to throughfall. This is the reason why throughfall does not age, as correctly pointed out by Referee1 a few paragraphs above. We made this clearer in the revised manuscript:

*"[. . .]. Only the same-time-step precipitation can contribute to throughfall in the ECH$_2$O model, whenever the resulting canopy storage would exceed the maximum canopy storage capacity (Maneta and Silverman, 2013), the latter being constant in our simulations. As a result, intercepted water eventually evaporates from the canopy and does not interact with the surface/subsurface. [. . .]"*

**P8L13 "PET" Please consider using Epot, as PET could also be precipitation, evaporation, and temperature.**

We realized that this acronym is not used anywhere else in the manuscript, so was

removed from the revised manuscript.

**P11L3-4 "Autumn" Lowercase letters**

It has been corrected in the revised manuscript

**P13L26-27 "model-to-data ratio of standard deviation and model-data Pearson's correlation factor". Please consider discussion the merits and pitfalls of using these evaluation metrics. See for example (Biondi et al., 2012) for review of different validation procedures that might be of relevance.**

We thank the Referee for this suggestion. As stated in our reply to General Comment 2, our approach consists in a generic evaluation of the new model using an ensemble of diverse isotopic datasets across ecohydrological compartments. This is why we rely on visual inspection (recommended in Biondi et al., 2012) as well as on generic metrics of model skill. The mean absolute error gives a generic quantification model-data fit across different type with lower sensitivity to high values within time series, contrary to metrics based on squared differences (such as RMSE or NSE; Krause et al., 2005; Legates and McCabe, 1999) and, to a lesser extent the Kling-Gupta Efficiency (KGE, Kling et al., 2012). In addition, NSE and KGE have been developed primarily for extracting information (and scores) from stream hydrograph for time series with a large number of points and pronounced variability, which is not the case for most isotopic datasets used here. Following the Referee's comment, in the revised manuscript we have used the mean absolute error (MAE) and Pearson's correlation factor as reference metrics. First, model-data MAE is shown for all relevant time series (displayed the ensemble median so as not to overload the figures). Second, Fig. 8 has been modified in order to display the normalized MAE (using the range of values of observations) against the Pearson's correlation factor:

*"**Figure 8.** Summary of model performance in the dual space of mean absolute error (normalized by the observed range of values) and Pearson's correlation factor between modelled and observed time series, for (a) $\delta^2 H$ and (b) lc-excess, showing the median*

*and 90%-spread over the ensemble. The size of each symbol is proportional to the logarithm of the number of observation points available. Performances in soil compartments at Forest site A are further separated between periods 2013 and 2015-2016 (the latter indicated with an asterisk), corresponding to two separate field data collection campaigns. Two groundwater wells are presents at the peat site."*

As pointed out in Biondi et al. (2012) and elsewhere, normalized MAE provides a more balanced evaluation, permits a direct comparison between different types of observables of varying distributions and dynamics, and is sensitive to model biases. The Pearson's correlation on the other hand captures very well if the model and observables have similar variances, but does not capture biases and is not robust to outliers, especially for time series with few points and/or low variability (groundwater and xylem). In response to this comment these edits have been brought to the main text in Sect 3.4 (P13L26):

*"As outlined in Sect. 1, our model evaluation is meant to test the ability of $ECH_2O$ to generically simulate isotope dynamics across compartments. We used mean absolute error (MAE) to quantify model-data fit for all isotopic outputs, some of which present low temporal variability, have skewed distributions, or have a relatively lower sampling record and resulting in typical hydrograph-oriented efficiency metrics (e.g., Nash-Sutcliffe or Kling-Gupta, Nash and Sutcliffe, 1970; Kling et al., 2012) being less applicable. The median value are shown on corresponding time series (Figs. 3–7). It is then normalized by each dataset range and used in conjunction with Pearson's correlation factor in Fig. 8 as a summary of model performance. The correlation coefficient axis in this dual model performance space represents the quality of the model in representing the variation of the data, while the normalized MAE axis provides information on the accuracy (bias) of the model."*

And in section 4.1 (P17L13):

*"A summary of model performance is shown in Fig. 8 for all sites/compartments, using*

*the dual space of normalized MAE (using each dataset range, x-axis) and Pearson's linear correlation factor (y-axis). The vast majority of median normalized MAE were below 1, and more than half of evaluated datasets showed values below 0.5. Values above 0.7 were mostly found for groundwater and xylem compartments, a clustering especially marked for $\delta^2 H$. In addition, most median model-data correlations were significantly positive between 0.4 and 0.85, noting a tighter clustering around high values for $\delta^2 H$ than lc-excess. Insignificant or negative correlations were mostly found where only a few data points were available (xylem) or where seasonal variability was low (e.g. groundwater). Interestingly, median model-data agreement in topsoil at Forest site A significantly differed between 2013 (mobile water sampling via lysimeters) and the 2015-2016 period (bulk water sampling via direct equilibration). This was notable in the dramatic increase of model-data correlation (0.17 to 0.8) and decrease of normalized MAE (0.5 to 0.25) for topsoil $\delta^2 H$ in the latter case, which is consistent with our interpretation that the simulated soil water composition represents that of bulk water."*

**P14 Sect 4.1. The time series section is detailed and provide considerable amount of information. However, it is also difficult for the reader to quickly get a grasp of the main strength and weaknesses of the model. Please consider including e.g., scatterplots.**

Fig. 8 of the revised manuscript provides the recommended scatterplots (see reply to previous referee comment). We have also edited the manuscript to facilitate the interpretation of the figure and guide the reader through the description of the evaluation metrics (at the very end of Sect. 3.4).

**P21 Fig 9, P23 Fig 10: Possibly consider moving some of the maps to the SI, and condense the information by grouping (by e.g., riparian/upstream/downstream etc types of regions).**

Please refer to our reply to General Comment 5.

**P26L9- "By keeping the. . ." Parts of this could also be modelling rationale that**

**could been useful in the introduction section or model set-up.**

Following this suggestion, we have emphasized this aspect in the abstract (see related comment above), and at the end of the introduction just after the list of research questions (also reformulated, see related comment above):

*"These questions are here addressed by testing the new tracer-enhanced model (ECH$_2$O-iso, Sect. 2) in a small, low-energy montane catchment (Sect. 3). This site has previously been modelled applying ECH$_2$O for calibration, using multiple datasets of long-term ecohydrological fluxes and storage variables (Kuppel et al., 2018). We take advantage of this earlier work as a reference ensemble of calibrated model parameterizations, and no additional isotopic calibration is conducted. In addition to using long-term, high resolution isotopic datasets for rainfall and runoff (2H and 18O), we assess the spatio-temporal variations of model-data agreement in soil water, groundwater, and plant xylem at different locations (Sect 4.1). Following this generic evaluation, the model is used to infer seasonally-varying patterns of water fluxes and isotopes signatures (Sect. 4.2), and water age (Sect. 4.3). Model strengths and weaknesses, insights in processes and potential ways forward are discussed in Sect. 5, before drawing conclusions in Sect. 6."*

**P3019: "ecohydrological feedbacks". A bit general, and not clear what the authors mean. Ecosystem response in terms of CO2 fertilisation and root depth development?**

The term "reciprocal ecohydrological feedbacks" here only encapsulates the reciprocal feedbacks between plant water use and terrestrial water pathways, in the face of environmental change. Given the simplified biogeochemistry used in ECH$_2$O, the effect of CO2 fertilization, of changes in nutrient availability, or of rooting depth development cannot be explored at present. We have modified this sentence in the revised manuscript as follows:

*"[. . .] Complementing more conceptual approaches, the physical basis of the ECH$_2$O-*

*iso model further provides the potential to extrapolate these insights beyond recorded conditions and scales, and to notably project the reciprocal feedbacks between plant water use, hydrological pathways and potential environmental changes."*

**References**

Berry, Z. C., Evaristo, J., Moore, G., Poca, M., Steppe, K., Verrot, L., Asbjornsen, H., Borma, L. S., Bretfeld, M., Hervé-Fernández, P., Seyfried, M., Schwendenmann, L., Sinacore, K., De Wispelaere, L., and McDonnell, J.: The two water worlds hypothesis: Addressing multiple working hypotheses and proposing a way forward, Ecohydrology, p. e1843, https://doi.org/10.1002/eco.1843, 2017.

Biondi, D., Freni, G., Iacobellis, V., Mascaro, G. and Montanari, A.: Validation of hydrological models: Conceptual basis, methodological approaches and a proposal for a code of practice, Phys. Chem. Earth, Parts A/B/C, 42–44, 70–76, doi:10.1016/j.pce.2011.07.037, 2012.

Birkel, C. and Soulsby, C.: Advancing tracer-aided rainfall–runoff modelling: a review of progress, problems and unrealised potential, Hydrol. Process., 29(25), 5227–5240, 2015.

Coenders-Gerrits, A. M. J., van der Ent, R. J., Bogaard, T. A., Wang-Erlandsson, L., Hrachowitz, M. and Savenije, H. H. G.: Uncertainties in transpiration estimates, Nature, 506(7478), E1–E2, doi:10.1038/nature12925, 2014.

Evaristo, J., Jasechko, S. and McDonnell, J. J.: Global separation of plant transpiration from groundwater and streamflow, Nature, 525(7567), 91–94, doi:10.1038/nature14983, 2015.

Good, S. P., Noone, D. and Bowen, G.: Hydrologic connectivity constrains partitioning of global terrestrial water fluxes, Science, 349(6244), 175–177,

doi:10.1126/science.aaa5931, 2015.

Jasechko, S., Sharp, Z. D., Gibson, J. J., Birks, S. J., Yi, Y. and Fawcett, P. J.: Terrestrial water fluxes dominated by transpiration, Nature, 496(7445), 347–50, doi:10.1038/nature11983, 2013.

Kling, H., Fuchs, M., Paulin, M.: Runoff conditions in the upper Danube basin under an ensemble of climate change scenarios. J. Hydrol. 424, 264–277, 2012.

Krause, P., Boyle, D.P., Bäse, F.: Comparison of different efficiency criteria for hydrological model assessment. Adv. Geosci. 5, 89–97, 2005.

Kuppel, S., Tetzlaff, D., Maneta, M. P., and Soulsby, C.: What can we learn from multi-data calibration of a process-based ecohydrological model?, Environmental Modelling Software, 101, 301–316, https://doi.org/10.1016/j.envsoft.2018.01.001, 2018.

Legates, D.R., McCabe, G.J.: Evaluating the use of "goodness-of-fit" measures in hydrologic and hydroclimatic model validation. Water Resour. Res. 35, 233–241, 1999.

Lozano-Parra, J., Maneta, M. P., and Schnabel, S.: Climate and topographic controls on simulated pasture production in a semiarid Mediterranean watershed with scattered tree cover, Hydrology and Earth System Sciences, 18, 1439, 2014.

McGuire, K. J. and McDonnell, J. J.: A review and evaluation of catchment transit time modeling, J. Hydrol., 330(3), 543–563, 2006.

Maneta, M. P. and Silverman, N. L.: A spatially distributed model to simulate water, energy, and vegetation dynamics using information from regional climate models, Earth Interactions, 17, 1–44, 2013.

Schlesinger, W. H. and Jasechko, S.: Transpiration in the global water cycle, Agric. For. Meteorol., 189–190, 115–117, doi:10.1016/j.agrformet.2014.01.011, 2014.

Simeone, C.: Coupled ecohydrology and plant hydraulics model predicts Ponderosa seedling mortality and lower treeline in the US Northern Rocky

Mountains, Master Thesis, University of Montana. [online] Available from: https://scholarworks.umt.edu/etd/11128, 2018.

Sprenger, M., Tetzlaff, D., Buttle, J., Laudon, H., Leistert, H., Mitchell, C. P., Snelgrove, J., Weiler, M. and Soulsby, C.: Measuring and Modeling Stable Isotopes of Mobile and Bulk Soil Water, Vadose Zone J., 17(1), 2018.

Troy, T. J., Pavao-Zuckerman, M. and Evans, T. P.: Debates-Perspectives on socio-hydrology: Socio-hydrologic modeling: Tradeoffs, hypothesis testing, and validation, Water Resour. Res., 51(6), 4806–4814, 2015.

Wei, Z., Yoshimura, K., Wang, L., Miralles, D. G., Jasechko, S. and Lee, X.: Revisiting the contribution of transpiration to global terrestrial evapotranspiration, Geophys. Res. Lett., 44(6), 2792–2801, doi:10.1002/2016GL072235, 2017.

Zhang, M., Liu, N., Harper, R., Li, Q., Liu, K., Wei, X., Ning, D., Hou, Y. and Liu, S.: A global review on hydrological responses to forest change across multiple spatial scales: Importance of scale, climate, forest type and hydrological regime, J. Hydrol., 546, 44–59, 2017.

**Calibration datasets**

**Stream discharge**
*Outlet*

**Soil moisture**
*Peat*

*Gley*

*Podzol*

*Forest site B*

**Pine transpiration**
*Forest site A*

*Forest site B*

**Net radiation**
*Valley bottom*

*Bog*

*Hilltop*

**Model evaluation**

02/2013    02/2014    02/2015    02/2016
    08/2013    08/2014    08/2015    08/2016

**Fig. 1.** New Figure S1.

Precipitation

Upstream cell(s)

Snowfall [1] Rainfall

Interception storage

[2a]

Canopy evaporation

[6] [2b]

Downstream cell

Snow throughfall

[4]

Liquid throughfall

[3]

Snowpack

Snowmelt [7]

Ponding

[12]*

Run-off

Run-on

[8]*

Infiltration [5a], [9a]*

Return flow [11d]

Run-off to stream

[12]°

Topsoil (layer 1)

Streamflow

Percolation [5b], [9b]*

Stream

Return flow [11c]

Streamflow

[8]°

[13]°

Intermediate (layer 2)

Seepage to channel

[9]°

Percolation [5c], [9c]*

Return flow [11b]

GW recharge [5d], [9d]*

GW inflow

[8]

Deeper soil (layer 3)

[11a]

Groundwater

Return flow

GW outflow

[10]

[5e] Seepage to bedrock

Soil evaporation

Transpiration

**Fig. 2.** New Figure 1

**a** δ²H

**b** lc-excess

Pearson's correlation factor

Normalized mean absolute error

Location: Forest A, Forest B, Heather A, Heather B
Podzol site, Gley site, Peat site, Outlet
Store: Topsoil layer, 2nd soil layer, Groundwater, Root uptake, Stream

**Fig. 3.** New Figure 8

---

## Author Comment (AC2) · 4 Jun 2018

The authors would like to thank the Referee 2 for her/his valuable comments and suggestions to strengthen the analysis presented in this manuscript. They have been taken into account in the revised manuscript, as follows (original referee's comments in bold):

**Kuppel et al. presents a physically-based ecohydrological model ECH$_2$O-iso that can track water isotopic tracers (2H and 18O) and age. The ECH$_2$O-iso is an extension of the ECH$_2$O model (Maneta and Silverman, 2013). The ECH$_2$O-iso model was evaluated at the Bruntland Burn catchment in the Scottish Highlands, and the simulation results show reasonable agreements with the isotopic mea-**

**surements. The paper is well written and structured, and it could be a potentially useful contribution to the literature. However, the authors used very general terms in many parts of their model evaluation, which makes it diffcult to assess the reliability of their results. For example, no statistics were shown on any of the time series plots, so there is no way that the readers can examine the model performance. Therefore, a major revision is suggested to improve the presentation of the current manuscript.**

We thank the reviewer for this suggestion. The mean absolute error (MAE) values have been added to all relevant time series in the revised manuscript (showing the ensemble median only, so as not to overload the figures), complementing the evaluation metrics provided in summary Fig. 8. The latter has according been modified, by using normalized MAE (using each datasets range) against the Pearson's correlation factor (the revised figure can be found at the end of this doucment):

*"**Figure 8.** Summary of model performance in the dual space of mean absolute error (normalized by the observed range of values) and Pearson's correlation factor between modelled and observed time series, for (a)$\delta^2 H$ and (b) lc-excess, showing the median and 90%-spread over the ensemble. The size of each symbol is proportional to the logarithm of the number of observation points available. Performances in soil compartments at Forest site A are further separated between periods 2013 and 2015-2016 (the latter indicated with an asterisk), corresponding to two separate field data collection campaigns. Two groundwater wells are presents at the peat site."*

Additionally, we added a justification for using these metrics at the end of Sect. 3.4 (P13L26):

*"As outlined in Sect. 1, our model evaluation is meant to test the ability of ECH$_2$O to generically simulate isotope dynamics across compartments. We used mean absolute error (MAE) to quantify model-data fit for all isotopic outputs, some of which present low temporal variability, have skewed distributions, or have a relatively lower*

*sampling record and resulting in typical hydrograph-oriented efficiency metrics (e.g., Nash-Sutcliffe or Kling-Gupta, Nash and Sutcliffe, 1970; Kling et al., 2012) being less applicable. The median value are shown on corresponding time series (Figs. 3–7). It is then normalized by each dataset range and used in conjunction with Pearson's correlation factor in Fig. 8 as a summary of model performance. The correlation coefficient axis in this dual model performance space represents the quality of the model in representing the variation of the data, while the normalized MAE axis provides information on the accuracy (bias) of the model."*

Finally, some descriptions of results have been made more precise, especially in the abstract and conclusions. Specific changes to the manuscript are detailed in 'specific comments'.

**Specific comments**

**Pg2, L9-12: The statement provided here seems not directly related to the paragraph above and below it. It is not clear what was the authors' attempt to deliver here. Also, what is "simplistic" meant by the authors with regard to the hydrology in land surface models?**

The phrasing was awkward and has been edited for clarity. Our intent was to indicate that land surface models are the only type of process-based models operating at scales larger than the hillslope where isotope tracking has been implemented (to our knowledge). However, simplifications in the representation of hydrological pathways, such as lateral connectivity, make them of limited applicability to understand water mixing and storage dynamics. In the revised manuscript, this paragraph has been reformulated as follows:

*"[. . .] While the simulation of energy budgets and biogeochemical cycles is increasingly*

*detailed in these land surface models –sometimes including vegetation dynamics–, the hydrology has, however, remained somewhat simplistic (or even absent) regarding lateral transfers as overland flow, shallow and deeper subsurface flows, and channel routing (Fan, 2015). This makes it difficult to take advantage of isotopes tracking to characterise the role of cascading downstream water redistribution in the spatial patterns of catchment functioning. [. . .]"*

**Pg3, L28-31: It is not clear how these questions being addressed in the paper. It would be very helpful if the authors could add more details about the experimental design to illustrate how these questions were linked to the results.**

We have modified the end of the introduction in the revised manuscript, so that the connection between research questions (now modified following Referee #1's suggestion) and our experimental/analytical design is clear (from P3L20 onwards):

*"This model was chosen because it provides a physically-based, yet computationally-efficient representation of energy-water-ecosystem couplings where intra-catchment connectivity (both vertical and lateral) can be explicitly resolved. In addition, $ECH_2O$ separately solves the energy balance at the top of the canopy and at the soil surface, allowing a process-based separation of Es, Et, and Ec. The novel isotopic and age tracking module is designed in a fashion directly consistent with the original model structure, assuming full mixing in each model compartment, and crucially without catchment-specific parameterization. The conceptualisation of evaporation fractionation uses the well-known Craig-Gordon approach (Craig and Gordon, 1965). We ask the following questions:*

- *To what extent can a hydrometrically-calibrated, physically-based hydrologic model correctly reproduce internal catchment dynamics of isotopes?*

- *What are the limitations of these isotopic simulations? Do they relate to the underlying model physics and/or to the tracking approach adopted?*

- *How useful and transferrable is this model framework for simulating spatio-
  temporal patterns of isotopes and water ages?*

*These questions are here addressed by testing this new tracer-enhanced model
(ECH$_2$O-iso, Sect. 2) in a small, low-energy montane catchment (Sect. 3). This site
has previously been modelled applying the original ECH$_2$O model for calibration, using
multiple datasets of long-term ecohydrological fluxes and storage variables (Kuppel et
al., 2018). We take advantage of this earlier work as a reference ensemble of cal-
ibrated model parameterizations, and no additional isotopic calibration is conducted.
In addition to using long-term, high resolution isotopic datasets for rainfall and runoff
($^2$H and $^{18}$O), we assess the spatio-temporal variations of model-data agreement in
soil water, groundwater, and plant xylem at different locations (Sect 4.1). Following
this generic evaluation, the model is used to infer seasonally-varying patterns of water
fluxes and isotopes signatures (Sect. 4.2), and water age (Sect. 4.3). Model strengths
and weaknesses, insights in processes and potential ways forward are discussed in
Sect. 5, before drawing conclusions in Sect. 6."*

**Pg4, L4: It might be better to change "climate" to "microclimate" since the spatial
and temporal scales used in the model is relatively small than the scales used in
climate science.**

Since the model is designed to be used at a range of spatial scales, including regional
studies (Simeone, 2018), in the revised manuscript we have used the term "local cli-
mate".

**Pg4, L11: What is the temperature threshold for the partitioning between liquid
and snow components? How does the model quantify snowpack depth for a
given amount of precipitating snow?**

For this threshold we use a default value of 2°C from Maneta and Silvermann (2013).
Snowpack depth is not quantified in the model; only the snow water equivalent is being
output by the model, and has been used for evaluation in Maneta and Silvermann

(2013). In the revised manuscript, the corresponding section of the paragraph now reads:

*"[...]. The capacity-excess P (i.e., throughfall) is partitioned between liquid and snow components using a snow-rain temperature threshold (fixed to 2°C) together with the minimum and maximum air temperature at each time step. [...]"*

**Pg4, L12: Canopy conductance is a key factor determining the amount of canopy transpiration. How is canopy conductance represented in the model? Is it simulated at each model time step?**

Stomatal conductance is represented by a Jarvis-type multiplicative model to account for the four major environmental stressors driving stomatal conductance, and then upscaled to canopy conductance using the leaf area index (LAI) (Maneta and Silverman, 2013):

$$g_{canopy} = g_{stoma}^{max} \cdot LAI \cdot f_{light} \cdot f_{temp} \cdot f_{VPD} \cdot f_{\Psi}.$$

Stomatal conductance is calculated for each vegetation type in each cell of the model. Here, $g_{stoma}^{max}$ is the maximum stomatal conductance (a calibrated parameter), while $f_{light}$, $f_{temp}$, $f_{VPD}$, and $f_{\Psi}$ are efficiency factors (range 0-1), respectively, which account for the effect of incoming shortwave radiation ($R_{SW\downarrow}$), air temperature ($T_a$), vapor pressure deficit at the leaf-air interface ($e_a^* - e_a$), and soil matric potential ($\Psi$). All these variables are calculated at each time step for each vegetation type present in the grid cells, noting that $f_{\Psi}$ is dynamically updated within the Newton-Rapson loop used to solve the 3-equations system for the canopy-level energy balance (see Appendix A1 in Kuppel et al., 2018 for further details):

$$f_{light} = \frac{R_{SW\downarrow}}{R_{SW\downarrow} + \phi_{SW\downarrow}}$$

$$f_{temp} = \left[\frac{T_a - T_{min}}{T_{opt} - T_{min}} \cdot \frac{T_{max} - T_a}{T_{max} - T_{opt}}\right]^{\left(\frac{T_{max} - T_{opt}}{T_{opt} - T_{min}}\right)}$$

$$f_{VPD} = \exp\left[-\phi_{e_a} \cdot (e_a^* - e_a)\right]$$

$$f_\Psi = \frac{1}{1 + \left(\frac{\Psi}{\Psi_d}\right)^c}$$

where $\phi_{SW\downarrow}$, $T_{min}$, $T_{opt}$, $T_{max}$, $\phi_{e_a}$, $\Psi_d$, and $c$ are empirical coefficients whose values are taken from the literature ($\phi_{SW\downarrow}$, $T_{min}$, $T_{max}$, $\phi_{e_a}$) or calibrated ($T_{opt}$, $\Psi_d$, and $c$). While adding this full description is beyond the scope of the paper, in the revised manuscript we modified the sentence highlighted for the Referee, as follows:

*"The canopy energy balance then separately yields plant transpiration ($E_t$) and evaporation of intercepted water ($E_c$). The calculation of $E_t$ uses, for each vegetation type, the canopy conductance at each time step based on a Jarvis-type multiplicative model accounting for environmental limitations of incoming solar radiation, $T_a$, vapor pressure deficit at the leaf surface, and soil water potential (see Maneta and Silverman (2013) and Appendices in Kuppel et al. (2018) for a more detailed description). Infiltration of surface water [. . .]"*

**Pg6, L3: $\Delta$t is redundant here as it has been defined right above eqn (2).**

This redundant definition has been removed.

**Pg8, L13: Could the authors provide any reference for the amount of PET estimated at the study site?**

We noticed that the value reported here is an estimate of actual evapotranspiration derived from applying the Penman-Monteith equation adjusted for heather shrub aerodynamic roughness (Birkel et al., 2011). This correction, reference, and precision has been added to the revised manuscript.

[Figure]

**Pg11, L27-29: Were there any missing data during the measurement period? If so, what was the gap-filling treatment for the meteorological observations? Also, what was the temporal resolution of the meteorological observations?**

The three weather stations at the catchment provided micrometeorological measurements at an original resolution of 15 minutes. Some measurements were sparsely missing in each of the stations records, with gaps ranging from one 15-min time step to a few days (notably during severe rainstorms at the beginning of 2016). There was however no instance of data simultaneously missing from all three stations, so that the daily inputs used for our simulations did not require a specific temporal gap-filling approach in the preprocessing stage. The revised manuscript includes information about the original temporal resolution of the raw meteorological data.

**Pg 12, L 13: How did the authors determine the transient dynamics has been removed after a 3-year spin up period?**

It was achieved by visual inspection of the time series of hydrometric and isotopic variables at the set of locations used in this study: through incrementing the spinup length starting from 1 to 6 years; no significant changes or trends were observed beyond 3 years of spinup. We added this precision in the revised manuscript (P12L11):

*"For all simulations a 3-year spin up period was added using the first three years of isotopic and climatic model inputs, as preliminary sensitivity tests combined with a visual inspection of simulated hydrometric and isotopic time series at the locations used in this study (Sect. 3.2) indicated it was sufficient to remove transient dynamics."*

**Pg 12, L21: Why did the authors set the depth of the first soil layer to 0.001 m? How sensitive does the model respond to the changes in the depth of the first soil layer?**

The depth of the first layer was set to 0.001 m at locations where a significant proportion (>0.5) of the grid cell area is bare soil, which always corresponds to locations with

exposed bare rock. This choice was made to limit the local soil evaporation simulated by the model (which only occurs in the first soil layer and thus is strongly controlled by its depth), and avoid producing an unrealistic degree of isotopic fractionation. Sensitivity tests shows that the overall effect was small in simulating the water balance given the relatively small area covered by exposed rock, and that the isotopic composition in downstream soils and in the stream channel was barely affected by this choice of a very thin topsoil. In the revised manuscript, the corresponding sentence has been modified as follows:

*"To avoid an overestimation of local soil evaporation and resulting isotopic fractionation in grid cells of exposed rock/scree, for simplicity we fixed the depth of the first soil layer to 0.001 m wherever the fraction of bare soil was larger than 0.5 – after performing a sensitivity analysis showing little effect on catchment water balance and downstream isotopic budgets."*

**Pg 13, L11: It should be Eq. 20 instead of Eq. 19.**

It has been corrected in the revised manuscript.

**Pg 19, L 26-27: How is the seasonal change of vegetation represented in the model? Was the increase of ecosystem transpiration resulted from the increase of vegetation leaf area or the increase of canopy conductance? Did the authors check the water loss from canopy evaporation? How much of difference did the model simulate between canopy evaporation and soil evaporation?**

For this study, we adopted the same configuration as Kuppel et al. (2018) where vegetation dynamics is turned off, i.e. leaf area index (LAI) remains constant. As a result, the variation in ecosystem transpiration results from that of canopy conductance (see our reply about its calculation a few comments above), and that of vapor gradient at the leaf surface (see also Eq. A4 in Kuppel et al. (2018)). We have added this precision in Sect. 3.3 (P12L20) of the revised manuscript:

*"As in Kuppel et al. (2018), the dynamic vegetation allocation module is switched off, so that leaf area index remains equal to initial values of 2.9, 1.6, 3.5, and 2 $m^2.m-2$ for Scots pines, heather shrubs, peat moss and grasslands, respectively (Albrektson, 1984; Calder et al., 1984; Bond-Lamberty and Gower, 2007; Moors et al., 1998)."*

The simulation ensemble provides a catchment-wide values of 354±50 mm/yr for canopy evaporation (here understood as interception losses plus transpiration), which is much higher than soil evaporation (59±22 mm/yr). Note that this large dominance of canopy evaporation ( 85-90% of the evaporative losses) over soil evaporation was also highlighted by observation-based, plot-scale studies at the same catchment in a Scots pine stand (Wang et al., 2017a) and at a heather plot (Wang et al., 2017b).

**Pg29, L23: Please change "T he" to The.**

It has been corrected in the revised manuscript.

**Pg30, L13-14: This is a very general statement. It would be very helpful if the authors could revise it with more specific terms so the readers can catch up easily.**

This is a good suggestion, and we provide more specific summary in the revised manuscript:

*"Evaluated against a multi-site, extensive isotopic dataset encompassing a wide range of ecohydrological compartments (soil moisture, groundwater, plant xylem, and stream water) across hydropedological units, the model has generically shown good performance in reproducing the seasonal and higher-frequency variations of absolute and relative isotopic content (ïĄď2H and lc-excess, respectively)."*

**Pg30, L14-15: Again, it is difficult for the readers to understand why this wound indicate the model is correct in both energy celerity and flow velocity viewpoints. It might be useful to explain what exactly are the energy celerity and flow velocity viewpoints meant by the authors.**

The definition of celerity and velocity viewpoints, given in the abstract and introduction, are here repeated for clarity in the revised manuscript:

*"This isotope-based evaluation suggests a correct capture of the velocity fields (i.e., how fast water parcels move) across the catchment, and complements a previous calibration and evaluation mostly using hydrometric observations (water fluxes and stores) which indicated a good simulation of catchment functioning from a celerity viewpoint (i.e., how fast energy propagates via the hydraulic gradient) (Kuppel et al., 2018). Satisfying this dual velocity-celerity perspective is key to characterizing water pathways [. . .]"*

**References**

Albrektson, A.: Sapwood basal area and needle mass of Scots pine (Pinus sylvestris L.) trees in central Sweden, Forestry, 57(1), 35–43, 1984.

Birkel, C., Tetzlaff, D., Dunn, S. M., and Soulsby, C.: Using time domain and geographic source tracers to conceptualize streamflow generation processes in lumped rainfall-runoff models, Water Resour. Res., 47, W02 515, https://doi.org/10.1029/2010WR009547, 2011.

Bond-Lamberty, B. and Gower, S. T.: Estimation of stand-level leaf area for boreal bryophytes, Oecologia, 151(4), 584–592, doi:10.1007/s00442-006-0619-5, 2007.

Calder, I. R., Hall, R. L., Harding, R. J. and Wright, I. R.: The use of a wet-surface weighing lysimeter system in rainfall interception studies of heather (Calluna vulgaris), J. Clim. Appl. Meteorol., 23(3), 461–473, 1984.

Craig, H. and Gordon, L. I.: Deuterium and oxygen 18 variations in the ocean and the marine atmosphere, Stable Isotopes in Oceanic Studies and Paleotemperatures, 1965.

Kuppel, S., Tetzlaff, D., Maneta, M. P., and Soulsby, C.: What can we learn from multi-data calibration of a process-based ecohydrological model?, Environmental Modelling Software, 101, 301–316, https://doi.org/10.1016/j.envsoft.2018.01.001, 2018.

Lozano-Parra, J., Maneta, M. P., and Schnabel, S.: Climate and topographic controls on simulated pasture production in a semiarid Mediterranean watershed with scattered tree cover, Hydrology and Earth System Sciences, 18, 1439, 2014.

Moors, E., Stricker, J. and van der Abeele, G.: Evapotranspiration of cut over bog covered by Molinia Caerulea, Wagenigen., 1998.

Maneta, M. P. and Silverman, N. L.: A spatially distributed model to simulate water, energy, and vegetation dynamics using information from regional climate models, Earth Interactions, 17, 1–44, 2013.

Simeone, C.: Coupled ecohydrology and plant hydraulics model predicts Ponderosa seedling mortality and lower treeline in the US Northern Rocky Mountains, Master Thesis, University of Montana. [online] Available from: https://scholarworks.umt.edu/etd/11128, 2018.

Wang, H., Tetzlaff, D., Dick, J.J., Soulsby, C.:. Assessing the environmental controls on Scots pine transpiration and the implications for water partitioning in a boreal headwater catchment. Agric. For. Meteorol. 240, 58e66, 2017a.

Wang, H., Tetzlaff, D., Soulsby, C.:. Testing the maximum entropy production approach for estimating evapotranspiration from closed canopy shrubland in a low-energy humid environment. Hydrol. Process., 31, 4613-4621, 2017b.
* * *
The plot, top panel labeled **a** δ²H, bottom panel labeled **b** lc-excess. Y-axis: Pearson's correlation factor. X-axis: Normalized mean absolute error.

Legend:

**Location**
- ○ Forest A
- □ Forest B
- ◇ Heather A
- △ Heather B
- ▽ Podzol site
- ⬦ Gley site
- ⊠ Peat site
- ⊗ Outlet

**Store**
- ● Topsoil layer
- ● 2$^{nd}$ soil layer
- ● Groundwater
- ● Root uptake
- ● Stream

**Fig. 1.** New Figure 8

---

## Author Comment (AC3) · 4 Jun 2018

The authors would like to thank the Referee 3 for her/his valuable comments to deepen the discussion of the conceptualisation adopted and the results. It has been taken into account in the revised manuscript and we reply point-by-point in the following (original referee's comments in bold).

**This review report is for the manuscript, entitled: "ECH$_2$O-iso 1.0: Water isotopes and age tracking in a process-based distributed ecohydrological model" by Kuppel et al.. This study embedded the water isotopic tracers and age into an ecohydrological model, ECH$_2$O and then applied this model onto a small catch-**

**ment. This model, therefore, could simulate the spatio-temporal variation of water flux and water isotopic composition in soil moisture, plant xylem, and groundwater. Overall speaking, I enjoyed reading this study which, indeed, is a great and innovative work. The spatio-temporal patterns of water isotopes can be demonstrated now and the hypothesis we have been concerned can be tested. The simulation is promising, which indicates that the present concepts and knowledge are tentatively correct. However, there are still some concerns that should be addressed for completing the statements.**

**First of all, this study simulated the hydrological processes without parameterization and calibration. Although the lack of calibration is a good way to test hypothesis comprehensively, it would lower the practical applicability for transferring this model to other catchments. This Aberdeen catchment with intensive observations is quite unique around the world. Therefore, it would be great to discuss the potential parameterization, particularly for the soil moisture, transpiration, and groundwater. The parameterization could not only increase the applicability for other catchments, but also help to introduce the landscape characteristics into the parameters, which is an important concern of critical zones where researchers attempt to incorporate the geophysical characterization into substance transport.**

We appreciate this comment. We must emphasize first that the ensemble of parameters sets used for the presented simulations derives from a multi-objective calibration conducted using hydrometrics and energy balance datasets as constraints (see Sect. 3.3), following the methodology of Kuppel et al. (2018). Most likely further calibration using isotopes datasets would introduce additional independent information capable of further refining the identification of model parameters. However, we chose not to conduct such calibration in order to put the new isotope tracking model to a fundamental test: we simply assess how the original ECH$_2$O structure (informed by hydrometry-based parameterization and successfully evaluated) performs when applying the current "tracer tracking" conceptualization.

This first step is in our view necessary to develop a solid and hypothesis-driven contribution to the emerging velocity-celerity (i.e., looking at both hydrological response and tracer transport) modelling community, even before engaging in the provision of a ready-to-use numerical tool. Although the positive results we present are very encouraging, our "minimalistic" approach also facilitates translating the model-data mismatches into specific development needs (as discussed in Sect. 5), something which would have been challenging otherwise, given the relative complexity of the original $ECH_2O$ model itself. We agree with the Referee that our catchment is unique in terms data availability. Hydrologists using this model in other catchments will mostly have hydrometry-related datasets available for calibration, with perhaps a few (if any) isotopic datasets. Assessing the information transferability from one viewpoint (energy celerity, provided by hydrometric datasets) to the other (water velocity as represented by isotopic composition and water ages), and their compatibility, is a reason why we did not include our isotopic datasets in the calibration.

We are nonetheless aware of the pressing need for tracer-enabled models such as $ECH_2O$-iso to retrieve landscape-relevant model parameterizations to leverage information-rich combinations of hydrometric and isotopic datasets. We are currently working on such a calibration approach using isotopes, along with further hypothesis-testing regarding soil mixing. We have added this aspect to the end of the revised abstract:

*"[...] Balancing the need for basic hypothesis testing with that of improved simulations of catchment dynamics for a range of applications (e.g., plant water use under changing environmental conditions, water quality issues, and calibration-derived estimates of landscape characteristics), further works could also benefit from including isotope-based calibration."*

**Secondly, the water isotopic measurement in soil moisture is very difficult and**

**tricky. As mentioned by Orlowski et al. (2016), it is intricate to determine the soil water isotopic composition. Presently, this model integrated all soil layers into one storage, which is acceptable, but can the authors explain more on what kind of soil water they simulated and what is their opinion about this issue in modeling work?**

**Finally, the observed lc-excess values of groundwater are higher than simulated ones indicating the exaggerated mixing across the soil profile. However, evaporation from shallow groundwater could raise the lc-excess variability as well. Can the authors explain more to this concern and provide some thinking for further modeling development?**

We grouped the two above comments by the Referee since they are interlinked.

Being able to compare simulated soil water isotopic composition with measurement representing a similar spatial footprint is key for correct model evaluation. Currently, the soil hydrology of ECH$_2$O differentiates between three vertical layers in each grid cell, (whose thicknesses are calibrated parameters). Our results present the soil water isotopic composition (Figs. 3–4) of the first two layers and correspond to bulk soil water. Although we mention it in the results and discussion section (P14L17, P18L10, and P26L23), this is missing from the method section. In the revised manuscript, we have added this precision in this isotopic model description (P6L9):

*"Note that because of its representation of a single, fully-mixed pool in each soil layer, ECH$_2$O-iso essentially provides a bulk water values for isotopic content and water ages. This needs to be kept in mind when comparing with soil isotopic datasets (see Sect. 3.2 and Sect. 4) and for the discussion (Sect. 5)."*

A significant contribution of the reported model-data lc-excess discrepancy can probably be attributed to the coarse vertical discretization of the soil profile (3 layers), which enhances mixing compared to approaches that use a finer discretization of the soil profile (e.g. Sprenger et al., 2018). Overestimated mixing may be a reason for the buffered

simulated isotopic signal and high lc-excess in the soil profile and in the groundwater.

This explanation is unsatisfying because it is rooted in the arbitrary numerical partitioning of the soil, and not on a hypothesis about hydrologic function. An alternative and more satisfying reason may be inadequacies of the full-mixing assumption and the need for a second type of water pool in each soil layer mixing at a different rate, which is a hypotheses guiding current model development. This dual mixing hypothesis relates to preferential flow pathways and is controlled by the degree of tension under which the water is held in the soil and the macro- to micro-scale variability of pore size (Beven and Germann, 2013). Despite being a long-standing issue in hydrological conceptualisation (Beven and Germann, 1982), associated efforts for catchment modelling are relatively rare and only recently gain momentum (e.g., Stump, 2007; Vogel et al., 2010; Sprenger et al., 2018; Smith et al., 2018). Without getting into the complexity (and potentially prohibitive computational cost) of applying a detailed description of a dual-porosity-based routing in the subsurface (e.g., Hutson and Wagenet, 1995) to the structure of the $ECH_2O$-iso model, we are currently exploring a parsimonious implementation for future studies with $ECH_2O$-iso. We have amended the corresponding part of Sect. 5.2 in the revised manuscript (P28L21):

*"[. . .] dynamics and tracer mixing (Beven and Germann, 2013). This would first involve implementing conceptualisation of micro-topographic controls on overland flow (Frei et al., 2010). Secondly, the significance of sub-surface dual pore space (matrix-macropore) representations of tracer flow paths and mixing has long been put forward (Beven and Germann, 1982) but modelling efforts relevant to catchment hydrology remain somewhat scarce (Stumpp et al., 2007; Stumpp and Maloszewski, 2010; Vogel et al., 2010; Sprenger et al., 2018; Smith et al., 2018). Bridging these detailed plot-to-hillslope-scale descriptions [. . .]"*

Finally, evaporation of shallow groundwater is not explicitly taken into account in the current $ECH_2O$-iso formulation of evaporative losses and isotopic fractionation. While these processes are not likely a major contributor to water fluxes and isotopic fractionation in our catchment (as hinted by the positive lc-excess values), future developments should take into account these process, which can become significant in locations with higher evaporative demand (e.g., Soylu et al., 2011).

**Thirdly, the simulated and observed deuterium composition and lc-excess in forest sites exist large discrepancies. It was straightforwardly attributed to the dependency among species. It indicated that vegetation pumping has great differences among species (e.g. heather and forest). It will be great if the authors can give some suggestions for further parameterization.**

The last paragraph of Sect. 5.2 (P28L19), discusses the observed model-data mismatch in Scot pine xylem and highlights limitations in our approach because: 1) we assumed soil-dependent root-profile, instead of a vegetation-dependent parameterization, and 2) unrepresented processes that could cause isotopic fractionation at different stage of xylem water cycling, e,g. during root uptake, via inner-stem exchange (e.g., xylem-phloem cycling) and via evaporation through the bark (see references in Sect. 5.2). These mechanisms are complex, non-exclusive, and the lack of a scientific consensus has made them a very active topic of ecophysiological research (Poca, *personal communication*). It is therefore difficult to suggest specific parameterization, but a first step to obtain probably requires to increase the temporal resolution of measurements and use it to derive a relationship that can be incorporated in models and that capture short-term variability (e.g., Martín-Gómez et al., 2016).

**References**

Beven, K. and Germann, P.: Macropores and water flow in soils, Water Resour. Res., 18(5), 1311–1325, 1982.

Beven, K. and Germann, P.: Macropores and water flow in soils revisited, Water Resour. Res., 49(6), 3071–3092, doi:10.1002/wrcr.20156, 2013.

Hutson, J. L. and Wagenet, R. J.: A multiregion model describing water flow and solute transport in heterogeneous soils, Soil Sci. Soc. Am. J., 59(3), 743–751, 1995.

Martín-Gómez, P., Serrano, L. and Ferrio, J. P.: Short-term dynamics of evaporative enrichment of xylem water in woody stems: implications for ecohydrology, Tree Physiol., 37(4), 511–522, 2016.

Smith, A. A., Tetzlaff, D. and Soulsby, C.: Using StorAge Selection functions to quantify ecohydrological controls on the time-variant age of evapotranspiration, soil water, and recharge, Hydrol Earth Syst Sci Discuss, 2018, 1–25, doi:10.5194/hess-2018-57, 2018.

Soylu, M. E., Istanbulluoglu, E., Lenters, J. D., and Wang, T.: Quantifying the impact of groundwater depth on evapotranspiration in a semi-arid grassland region, Hydrol. Earth Syst. Sci., 15, 787-806, https://doi.org/10.5194/hess-15-787-2011, 2011.

Sprenger, M., Tetzlaff, D., Buttle, J., Laudon, H., Leistert, H., Mitchell, C. P., Snelgrove, J., Weiler, M. and Soulsby, C.: Measuring and Modeling Stable Isotopes of Mobile and Bulk Soil Water, Vadose Zone J., 17(1), 2018.

Stumpp, C. and Maloszewski, P.: Quantification of preferential flow and flow heterogeneities in an unsaturated soil planted with different crops using the environmental isotope $\delta$18O, Journal of Hydrology, 394, 407–415, 2010.

Stumpp, C., Maloszewski, P., Stichler, W., and Maciejewski, S.: Quantification of the heterogeneity of the unsaturated zone based on environ mental deuterium observed in lysimeter experiments, Hydrological Sciences Journal, 52, 748–762, 2007.

Vogel, T., Sanda, M., Dusek, J., Dohnal, M., and Votrubova, J.: Using Oxygen-18 to Study the Role of Preferential Flow in the Formation of Hillslope Runoff, Vadose Zone Journal, 9, 252–259, https://doi.org/10.2136/vzj2009.0066, 2010.